

# Assessment of the physical vulnerability of buildings affected by slow-moving landslides

Qin Chen[1], Lixia Chen[2*], Lei Gui[1], Kunlong Yin[1], Dhruba Pikha Shrestha[3], Juan Du[4], Xuelian Cao[2]

[1]Engineering Faculty, China University of Geosciences, Wuhan, 430074, China

[2]Institute of Geophysics and Geomatics, China University of Geosciences, Wuhan, 430074, China

[3]Department of Earth Systems Analysis, Faculty of Geo-Information Science and Earth Observation (ITC), University of Twente, Enschede, 7500 AE, the Netherlands

[4]Three Gorges Research Center for geo-hazard, Ministry of Education, China University of Geosciences, Wuhan, 430074, China

*Correspondence to: Lixia Chen (lixiachen@cug.edu.cn)

**Abstract:** Physical vulnerability is a difficult fundamental issue in the risk assessment of slow-moving landslides. We aim to develop a method to analyze the physical vulnerability of buildings affected by slow-moving landslides. We calculate the landslide residual force on the buildings' foundation and the landslide safety factor where the buildings are located using the GEOSTUDIO code and landslide residual thrust method. Further, using

Timoshenko's deep beam theory, we analyze the physical response of buildings to understand potential inclination. By applying the modified Weibull function, we fit the physical vulnerability function based on the relation between damage degree and landslide intensity. We simulate three rainfall scenarios by employing the Pearson type III distribution model to evaluate changes in the landslide's residual thrust and corresponding buildings' damage degree. To obtain the contributions of the buildings' characteristics to physical vulnerability,

we conduct sensitivity analysis, demonstrating that the building length, foundation depth, and building width are the most critical factors. Two physical vulnerability curve sets are separately generated for four building lengths and five building foundation depths. The proposed method can be applied to establish the physical vulnerability of landslides. The established physical vulnerability curves are used for the quantitative risk assessment of slow-moving landslides.

**Keywords:** slow-moving landslides; physical vulnerability; building; vulnerability curves; risk

## 1 Introduction

Physical vulnerability is a fundamental and indispensable item in the risk definition presented by Varnes (1984). It can be defined as the degree of loss to a given element or set of elements within an area affected by a hazard (UNDRO, 1984). Physical vulnerability is measured on a continuous scale ranging from 0 (no loss) to 1 (total loss). For quantifying physical





loss, such as the structural damage, the physical vulnerability of the elements at risk can be achieved by assessing the

damage degree, resulting from the occurrence of a landslide of a given type and volume (Van Westen et al., 2006).

Recently, physical vulnerability is still a challenge, and there has been a growing interest in quantifying natural hazard

risk (Van Westen et al., 2006). To quickly and easily analyze the physical vulnerability, researchers have developed various

types of tools or software such as HAZUS-MH (FEMA, 2003), RiskScape (King and Bell, 2005), ARMAGEDOM (Sedan et

al., 2013), and CAPRA (https://ecapra.org/). HAZUS-MH (FEMA, 2003) is considered to be the initially introduced and the

most popularly applied software. RiskScape is a national-scale multi-hazard impact model in New Zealand, and

ARMAGEDOM is a tool for seismic risk assessment that has three different precision levels (regional territorial scale,

district-scale, and the district-scale with more detailed hazard description and physical vulnerability estimation). Majority of

the software are employed to analyze the physical vulnerability of earthquakes or multi-hazards, and very few can be utilized

for landslide hazard assessment. To solve this problem, Papathoma–Köhle et al. (2015) developed an integrated toolbox

designed for buildings subjected to landslides.

Meanwhile, signs of progress have been made in the past decade in landslide physical vulnerability assessment, and the

following four main approaches exist: expert judgment approach (Sterlacchini et al., 2007; Winter et al., 2014; Godfrey et al.,

2015; Guillard–Goncalves et al., 2016); statistical method approach (Ciurean et al., 2013; Ciurean et al., 2017); mechanics-

based approach (Luna et al., 2014; Liang and Xiong, 2019; Nicodemo et al., 2020); and integrated approach (Li et al., 2010;

Uzielli et al., 2015b). The results of these approaches include matrices, indicators, and fragility or physical vulnerability

curves or functions. For example, by utilizing the procedures motivated by the seismic risk analysis, Negulescu and Foerster

(2010) introduced a simplified methodology to evaluate the mechanical performances of buildings subjected to landslide

hazard. In addition, Totschnig et al. (2011) presented physical vulnerability curves for debris flow and torrent hazards. Wu et

al. (2011) constructed physical vulnerability curves for landslides by considering the landslides' impact energy and impact

impulse as the intensity indicators. By utilizing FLO-2D (a hydrologic and hydraulic modeling software of debris flow

propagation), Luna et al. (2014) discussed the physical vulnerability functions of buildings at debris flow risk. By integrating

the assessment of landslide intensity and buildings' resilience, Uzielli et al. (2015b) proposed a modified the physical

vulnerability function based on the one proposed by Li et al. (2010). Papathoma–Köhle (2015) related hazard intensity

(debris-flow depth) with the loss of buildings' damage to buildings' physical vulnerability curves. Soldato et al. (2017)

studied the empirical physical vulnerability curves for buildings by considering the debris-flow depth, the flow velocity, and

the impact pressure. Mavrouli et al. (2017) quantified the masonry-buildings' damage induced by rockfalls by calculating the

impact force of falling rocks on masonry buildings.





By employing indicators and constructing curves, this study proves that quantifying building damage is a possible

method for assessing the physical vulnerability. In addition, the slow-moving landslides are particular types of landslides

with slow velocity based on the velocity scale provided by Cruden and Varnes (1996). Slow-moving landslides on the pre-

existing sliding surfaces can cause differential settlement or tilt on structures. People are not usually endangered but and

damage to buildings and infrastructures may be high (Douglas, 2007). Slow-moving landslides are observed worldwide in

many countries, e.g., Italy (Cascini et al., 2008; Antronico et al., 2015; Uzielli et al., 2015a; Nicodemo et al., 2017; Borrelli

et al., 2018; Ferlisi et al.,2019), Canada (Clifton et al., 1986; Brooker and Peck, 1993; Moore et al., 2006; Barlow, 2000),

China (Chen et al., 2016; Zhang et al., 2018; Dong et al., 2018; Wang et al., 2018), USA (Esser, 2000), and Australia

(Jworchan et al., 2008).

Meanwhile, the approaches for assessing the physical vulnerability of slow-moving landslides are still limited. Because

slow-moving landslides have different intensity indicators and different destructiveness with debris flows, rockfalls, or fast-

moving landslides, the aforementioned approaches are not suitable for slow-moving landslides. Fell et al. (2008)

recommended that the physical vulnerability of elements at risk should be estimated for various landslide types. For the

slow-moving landslides, buildings' damage is more likely or even much more on some sensitive areas of the landslides

regardless of the total landslide displacement or the released energy such as the boundary or local scarps (Fell et al., 2008).

Therefore, this study focuses on the buildings' damage and the physical vulnerability located within the slow-moving

landslides.

The analysis of the performance of buildings during the landslide processes and considering the inventory of the

observed damage is a feasible methodology (Faella and Nigro, 2003). To investigate the physical vulnerability of the

buildings impacted by landslides, numerous studies have been previously conducted regarding the acquisition of landslide

deformation displacement or finding the statistical relation between the damage degree of buildings and landslide intensity

(Mansour et al., 2011; Abdulwahid and Pradhan, 2017; Nicodemo et al., 2017; Peduto et al., 2017; Chen et al., 2016; Peduto

et al., 2018). For example, Mansour et al. (2011) statistically investigated the relation between the movement and the

expected extent of damage to urban settlements. Based on the persistent scatterer interferometry, Lu et al. (2014) obtained

the slow-moving landslides velocity for estimating buildings' economic loss risk with a total affected area of more than 800

km$^2$. Ferlisi et al. (2015) reported that combining the differential interferometry (DInSAR) data and the results of

supplementary damage surveys on the slow-moving landslides allowed the preliminary generation of a (maximum velocity)

cause-effect (damage) relation. Peduto et al. (2017) applied landslide deformation (cumulative surface displacement and

differential settlement) as the input variables to construct the empirical fragility and physical vulnerability curves for

buildings. Further, detailed researches should be conducted for physical vulnerability by performing mechanical analysis on


buildings. By applying the horizontal strains and angular distortions to the numerical model, Infante et al. (2016) recently

generated physical vulnerability domains for buildings. Nicodemo et al. (2020) employed the equivalent frame method to

analyze the damage of a representative building in case of a slow-moving landslide by numerical modeling.

This study proposes a new method for assessing the physical vulnerability from the perspective of mechanics and

obtains its changes during the slow-moving landslides processes. We first calculate the thrust force of landslide acting on the

buildings' foundation and then analyze the buildings' physical response. Multi-scenarios were applied to help in constructing

the physical vulnerability curves. After the validation by utilizing an application on a typical building impacted by slow-

moving landslides, a sensitivity analysis was finally conducted on the parameters of the building and its foundation.

## 2 Proposed method

### 2.1 Force acting on the building foundation during the landslide process

To quantitatively evaluate the building's damage physical vulnerability during the landslide process, it is essential to

calculate the force acting on the building's foundation. In this study, landslide residual thrust force is calculated by

employing the residual thrust method, which is an original method for slope stability analysis that is extensively applied in

China (Nie et al., 2004). A slide-mass is divided into slices in this method. A force analysis is performed on each slice.

Therefore, we can easily obtain the thrust of a landslide by utilizing the arbitrary shape of the sliding surface and under

complex loads. The landslide residual force can be calculated by applying Eq. (1)–(6). What we need to examine is that

groundwater seepage should be considered under rainy conditions, which can be performed using the SEEP/W code

(GEOSTUDIO). Meanwhile, landslide safety factor can be calculated and used as the input of landslide intensity for

generating the physical vulnerability curve. Landslides with smaller safety factors are more unstable, resulting in greater

residual thrust on the building's foundation.

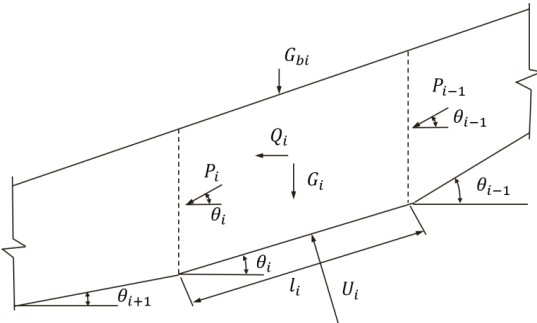

**Fig. 1. Computing model of residual thrust method with a broken-line slip surface.**

The safety factor of landslide, $F_s$, is defined as follows:



$$F_s = \frac{\sum_{i=1}^{n-1}\left(R_i \prod_{j=1}^{n-1} \psi_j\right) + R_n}{\sum_{i=1}^{n-1}\left(T_i \prod_{j=1}^{n-1} \psi_j\right) + T_n},\tag{1}$$

For a single slice, the residual thrust force of ith slice is given as follows:

$$P_i = P_{i-1} \times \psi_{i-1} + T_i - R_i/F_s,\tag{2}$$

$$F_i = P_i \times cos\theta_i,\tag{3}$$

$$R_i = [(G_i + G_{bi})sin\theta_i - Q_i sin\theta_i - U_i] \times tan\varphi_i + c_i l_i,\tag{4}$$

$$T_i = (G_i + G_{bi}) \times sin\theta_i + Q_i \times cos\theta_i,\tag{5}$$

$$\psi_{i-1} = cos(\theta_{i-1} - \theta_i) - sin(\theta_{i-1} - \theta_i)tan\varphi_i/F_s,\tag{6}$$

where $R_i$ denotes the resistance force of $i$th slice (KN/m), $T_i$ denotes the driving force of $i$th slice (KN/m), $P_i$ denotes the residual thrust of $i$th slice (KN/m), $\psi_i$ denotes the transmitting coefficient of $i$th slice, $G_i$ denotes the weight of $i$th slice (KN/m), $G_{bi}$ denotes the accessional vertical load of $i$th slice (KN/m), $\theta_i$ denotes the angle between the sliding surface and horizontal plane of the ith slice, $l_i$ denotes the length of $i$th slice (m), $c_i$ denotes the cohesion of $i$th slice (Kpa), $\varphi_i$ denotes the internal friction angle of ith slice, $U_i$ denotes the pore water pressure of $i$th slice (KN/m), $Q_i$ denotes the horizontal seismic force of $i$th slice, and $F_i$ denotes the horizontal component of landslide thrust (shown in Fig. 2).

The transformation of landslide residual thrust force on buildings' foundation depends on the distribution of force. According to Chinese standard (China Railway Second Survey and Design Institute, 1983) and Dai (2002), landslide thrust distribution is approximately assumed to be triangular, rectangular, or parabola shapes, based on the type of sliding mass materials. Each type of thrust distribution corresponds to a distribution function (Table 1).

**Table 1. Distribution functions of landslide thrust for various sliding mass materials of landslide.**

| Soil types | Distribution form (referred to China Railway Second Survey and Design Institute (1983) ) | Distribution functions (Referred to Dai(2002) ) |
|---|---|---|
| Clay, Soil-rock, Rock | Rectangle or parallelogram | $q(z) = \dfrac{F}{h}$ |
| Sand | Triangle | $q(z) = \dfrac{2F}{h^2}z$ |
| Between clay and sand | Parabola shape | $q(z) = \dfrac{1.8F}{h^2}z + \dfrac{F}{10h}$ |

Note: $F$ denotes the horizontal component of landslide residual thrust ($P_i$) in Eq. (3), and $h$ denotes the vertical distance from sliding surface to the ground surface (Fig. 2).




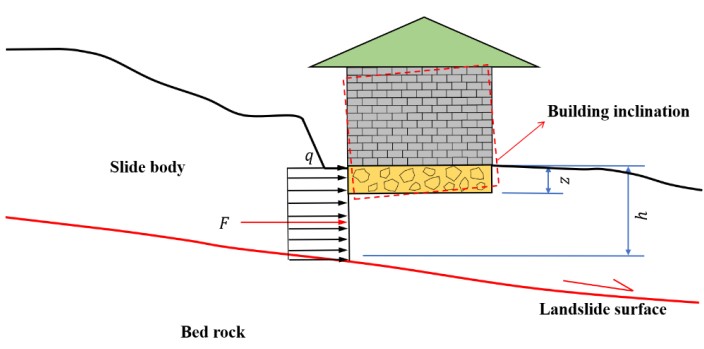

**Fig. 2. A schematic diagram of landslide thrust action on a building.**

## 2.2 Physical response of buildings

### 2.2.1 Inclination of buildings

The foundation of the masonry building affected by the landslide thrust can be simplified as a beam (Fig. 3). It has been

observed that real structures are normally very complicated, but the simplification of the beam helps in illustrating several

important features (Burland, 1974).

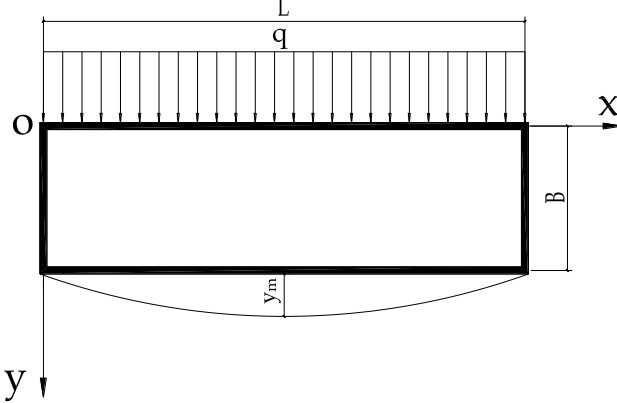

**Fig. 3. The simple beam with its foundation affected by landslide thrust.**

For illustrative purposes, we only consider the case of a beam with a uniform load. Timoshenko (1984) gave the

function of deflection for the uniform loaded beam of unit thickness flexing in both shear and bending as follows:

$$y(x) = \frac{qx}{24EI}\left(\frac{x}{L}\right)\left(\frac{x^3}{L^3} - 2\frac{x^2}{L^3} + 1\right) + \frac{3qL^2}{4GA}\left(\frac{x}{L}\right)\left(1 - \frac{x}{L}\right), \tag{7}$$

where $q$ denotes the distribution force on the foundation (KN/m), $L$ denotes the length of the building, $I$ denotes the moment

of inertia defined by $I = \frac{dW^3}{12}$, in which $d$ denotes the depth of the foundation and $W$ denotes the width of the building. In

addition, $E$ and $G$ denote Young's modulus and shear modulus of the foundation materials, respectively.

When $x = \frac{L}{2}$, the equation for the total central deflection is the following:





$$y_m = \frac{5qL^4}{384EI} + \frac{3qL^2}{16GA},\tag{8}$$

where the maximum deformation of the foundation is denoted by $y_m$. The following is the equation for the inclination of the building:

$$i = tan\alpha = \frac{y_m}{H_g} = \frac{1}{H_g}\left(\frac{5qL^4}{384EI} + \frac{3qL^2}{16GA}\right),\tag{9}$$

where $i$ denotes the inclination of the building, which is the ratio of the maximum deformation $y_m$ and the height of the

building calculated from the outdoor ground $H_g$.

### 2.2.2 Damage degree definition

In this study, the inclination ratio of a building is seemed as the damage degree, which is the ratio of the building's inclination to the threshold value. The damage degree is regarded as the output of the physical vulnerability. The degree of the building damage can be evaluated by utilizing some parameters, such as cracks in walls, inclination ratio, and the ratio of

maintenance cost and the original value of building (Alexander, 1986; Chiocchio et al., 1997; Cooper, 2008). Since cracks on walls are not visible, especially when the building with high stiffness is exceedingly inclined because of the ground deformation, they usually serve as the indicators of damage degree evaluation if the building stiffness is small (Finno et al., 2005). Therefore, the crack width is not the only indicator for building damage evaluation. Regarding inclination, it has been indicated that a visible deviation of members would often cause subjective feelings that were unpleasant and possibly

alarming (Burland, 1977). Therefore, the inclination has been chosen to represent the deformation of buildings (Huang, 2015).

Moreover, the integral inclination of the building is easy to measure. The standard for dangerous building appraisal (JGJ125-2016 China) provides the threshold value of inclination of single- or multi-story buildings (Table 2). Buildings with inclination exceeding 1/100 are considered to be dangerous.


**Table 2. The threshold value of building inclination (Ministry of Housing and Urban–Rural Development of PRC, 2016).**

| Height ( m ) | $H_g \leq 24$ | $24 < H_g \leq 60$ | $60 < H_g \leq 100$ |
|---|---|---|---|
| Threshold value $i_m$ | 1% | 0.7% | 0.5% |

Here, $H_g$ denotes the building height which is calculated from the outdoor ground.

By comparing the inclination of the building with the threshold value, the damage degree (*V*) can be calculated as follows:

$$V = \begin{cases} \frac{i}{i_m} = \frac{1}{H_g i_m}\left(\frac{5qL^4}{384EI} + \frac{3qL^2}{16GA}\right) & (i < i_m) \\ 1.0 & (i \geq i_m) \end{cases}.\tag{10}$$





The damage degree (*V*) ranges from 0 to 1.0. A higher value of *V* indicates that the damage degree is very close to the limited damage condition and more serious damage has occurred Equation (10) demonstrates that the building's inclination depends on the following three parameters of the building: size, material, and foundation depth. To ascertain the parameter with the highest significant impact on the degree of building damage, we can conduct a sensitivity analysis on these

parameters by employing the principle of controlling variables.

**2.3 Physical vulnerability function for masonry buildings**

**2.3.1 General functions**

In this study, we obtain the physical vulnerability curve by relating building damage degree with landslide safety factor (FS). It is important to indicate that FS is calculated for only the area where the target building is located, but not for the whole

landslide. Landslide intensity is directly proportional to its stability situation. A higher intensity corresponds to a higher thrust force on the building foundation and lower landslide safety factor. Thus, we utilize the reciprocal value of FS to be the landslide intensity in this study.

The relationship between building damage degree and the landslide intensity was fitted by employing Weibull (1951) function that produces an S-shaped curve. This type of distribution curve has been proved to be the best for physical

vulnerability analysis by Papathoma–Köhle et al. (2015). Based on these findings, a modified Weibull function for calculating physical vulnerability is defined as follows:

$$V = 1 - e^{-a\left(\frac{1}{Fs}\right)^b},$$   (11)

where *V* denotes physical vulnerability which is calculated by employing Eq. (10) in section 2.2.2; *Fs* is calculated by employing eq. (1) in section 2.1; *a* and *b* are constants to be determined.

**2.3.2 Determination of constants by applying multiple scenarios**

To determine the constants *a* and *b* in Eq. (11), we first obtain two or more scenarios, which can reflect the landslide safety factor and the building damage degree. Using several triggering scenarios, such as rainfall, earthquake, and reservoir water level fluctuation, we can obtain several safety factors, the corresponding landslide force on building foundation, and the building damage degree. Then, we apply the least-square method to obtain the constants based on the presupposed function

in Eq. (11).

In this study, rainfall is the key triggering factor for the landslide. Thus, we obtain rainfall scenarios by analyzing the precipitation using different return periods. Pearson type (PT) III distribution model (Lei et al., 2018; Radwan, Alazba, and Mossad, 2019) is applied because it is useful in rainfall-induced landslides; its probability density function is defined as follows:





$$f(x) = \frac{\beta^{\alpha}}{\Gamma(\alpha)}(x - a_0)^{\alpha-1}e^{-\beta(x-a_0)},$$
(12)

where parameters $\alpha, \beta, a_0$, can be given by the following three statistical parameters after conversion: $(\acute{x}, C_v, C_s)$. Thus, we

have that

$$
\begin{aligned}
\alpha &= \frac{4}{C_s^2} \\
\beta &= \frac{2}{\acute{x}C_vC_s} \\
a_0 &= \acute{x}\left(1 - \frac{C_v}{C_s}\right)
\end{aligned}
$$
(13)

where $\acute{x}$ denotes the average value, $C_v$ denotes the coefficient of variation, and $C_s$ denotes the coefficient of skewness.

From Eq. (13), the PT III distribution model has three undetermined parameters: $\acute{x}, C_v, C_s$. The principle of maximum

entropy, the methods of moments, and maximum likelihood estimation are employed to estimate the parameters for the PT

III distribution (Singh and Singh, 1988). We plot the physical vulnerability curve after obtaining the values of these three

parameters determined by different rainfall scenarios with varying return periods.

**3 Application of the proposed method**

**3.1 Geological settings and deformation of landslide**

The landslide called Manjiapo located in Sangzhi County, Zhangjiajie, China was selected as the case study (Fig. 4). The

area, which has an elevation ranging from 154 m a.s.l to 1890 m a.s.l, where Manjiapo landslide exists, is mountainous and

hilly. The climate of the area was humid subtropical, while the average annual rainfall was over 1400 mm.

The landslide covers an area of about $6.6 \times 10^4$ m$^2$ with an average thickness of 6.9 m and the estimated volume of 45.5

$\times 10^4$ m$^3$. It demonstrates strip shape in a plan with a longitudinal dimension of about 560 m and the average width of

approximately 176 m along the northwest (NW)–southeast (SE) direction. The elevation of the main crack is about 370 m

a.s.l. The toe of the landslide is located in the stream of the NW side, with an elevation of 272 m a.s.l.

The topography demonstrates a multi-step shape, and the height of most steps ranges from 1 to 3 m. The middle and

upper parts of the landslide are relatively gentle with a slope of about 8 °, while the lower part is steeper with a slope of

about 12 °. The sliding direction of the landslide includes two parts: the upper part orients at 335 °, and the lower part orients

at 313 °.

The main materials of the landslide comprise loos debris from silty clay and siltstone, in which the latter only

distributes in the middle and upper sections of the landslide (Fig. 5). The bedrock is argillaceous siltstone with a slope angle

of approximately 10 °. Based on the detailed landslide report provided by the China Geological Survey (Hunan Institute of




Xiangxi Geological Engineering Survey) in 2017, the shear-strength parameters of the slip soil of the landslide are presented

in Table 3. All the parameters were obtained by numerous sets of soil samplings on the landslide and laboratory tests.

**Table 3. Shear-strength parameters of Manjiapo landslide slip soils.**

|  | Dry condition | | Saturated condition | |
|---|---|---|---|---|
|  | $c/kpa$ | $\varphi/(°)$ | $c/kpa$ | $\varphi/(°)$ |
| Average | 11.98 | 9.09 | 5.85 | 6.84 |
| Variance | 1.56 | 2.25 | 0.79 | 0.64 |

Note: c denotes cohesion; $\varphi$ denotes friction angle; the permeability coefficient is 0.3, while the volume of water content is

0.4.

Manjiapo landslide has a history of 10-year displacement. Based on the descriptions of the residents, the landslide

occurred in August 2008, and there were a few ground fissures. Eight years later, heavy rainfall from 28th to 30th June

induced severe deformation of the landslide. In July 2017, the authors performed field investigation and observed that the

deformation mainly concentrated in the middle and upper parts of the landslide (Figs.4 and 6). Numerous tension cracks in

the upper part had a visible depth of 2–5 cm, with a length of 1 600 to 6 600 cm and the width of about 15 cm (Figs. 6a, b,

and c). In the middle part of the landslide, staggered extrusion deformation can be observed locally except for numerous

tension cracks.

Additionally, the soil permeability increases based on the influence of surface deformation that caused higher

groundwater level in silty clay layer. The shear strength of the soil mass decreased due to the softening of groundwater,

forming the sliding zone exposed by boreholes implemented in 2017. On the lower part of the landslide, the cracking of the

road was observed, and there is a little uplift deformation in the ground (Fig. 6(d)).

Rainfall was the most important triggering factor of Manjiapo landslide that underwent slow movement. Simple

measurements of the cracks and observation of the macroscopic deformation were performed on the landslide since 2016.

Based on the monitoring results, only heavy rainfall could cause the movement of the landslide. According to the

investigation performed on the borehole, the groundwater table is stable in dry seasons. Meanwhile, since the extreme

rainfall events were recently rare, the deformation of the landslide did not obviously change, which was similar to the

deformation situation in June 2016. For example, the cracks on the landslide did not expand, and the number of new cracks

was very few.





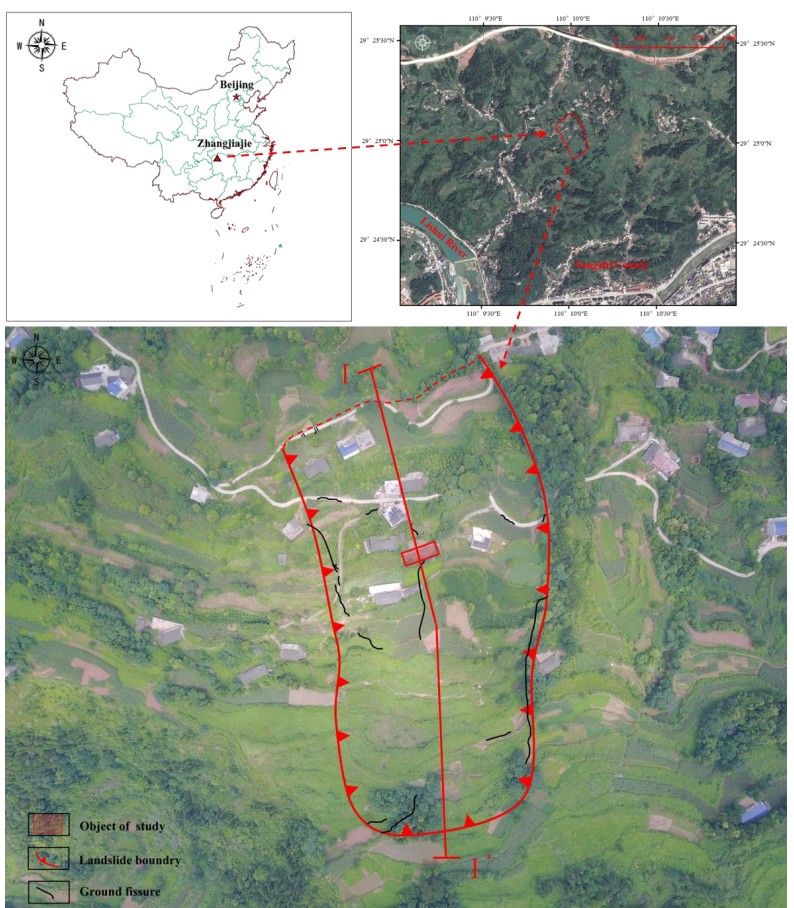

**Fig. 4. Location of the Manjiapo landslide. The Map of China is download from http://www.geodata.cn by National Earth System**
**Science Data Center, National Science & Technology Infrastructure of China. The map on the top right corner is produced from**
**Google Earth. The map showing the landslide boundary is from Unmanned Air Vehicle operated by the author groups in the field.**

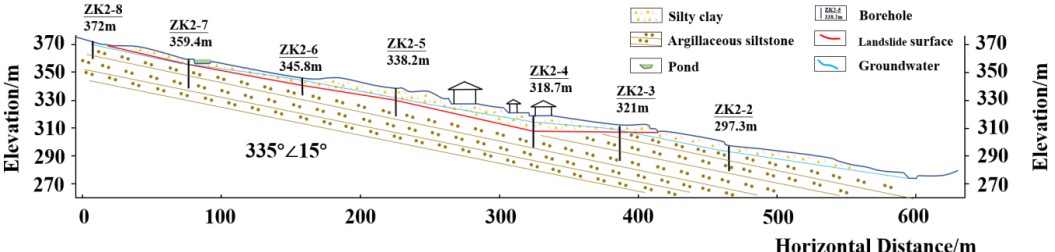

**Fig. 5. Geological profile of I-I' of the Manjiapo landslide (1:1 000).**



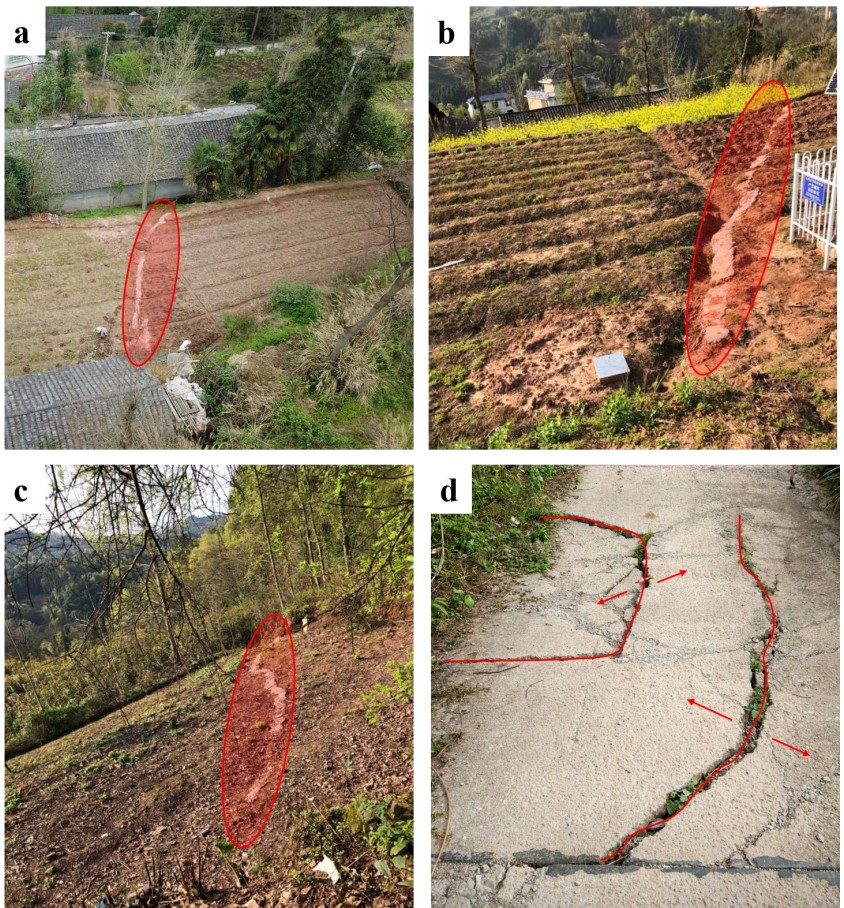

**Fig. 6. Ground cracks of Manjiapo landslide: (a) the middle part, (b) the upper part, (c) the right boundary, (d) the lower part.**

**3.2 Damaged buildings on the landslide**

In July 2017, we surveyed buildings on the Manjiapo landslide. There were 15 houses in the area affected by the landslide, in which 5 and 10 are brick-wood and brick-concrete buildings, respectively (Fig. 7). The buildings located in the middle part were the most severely damaged. Due to landslide deformation, the walls of these buildings were cracked and inclined. We

selected a building as the target object that experienced integral decline and severe cracks on the walls, and finally, it was banned due to being severely damaged.

The target building is a story masonry building with a length and a width of 25 m and 9 m, respectively. There were six rooms in the building, and each room was damaged from June 28th to 30th, 2016. The large-scale ground collapse occurred in rooms C,D, and E (Fig. 8). Meanwhile, the walls of these rooms developed numerous diagonal cracks with the width

from 2 cm to 8 cm. From the appearance of the building, the walls were heavily tilted, with inclination from 0.7% to 1.0% based on the measurement (Fig. 9).





**Table 4. Parameters of the building on Manjiapo landslide**

| Length $L(m)$ | Width $W(m)$ | Height $H_g(m)$ | Depth of foundation $d(m)$ | Young's modulus of Rubble masonry $E(Mpa)$ | E/G | Soil depth where the building located(m) |
|---|---|---|---|---|---|---|
| 25 | 9 | 2.8 | 1 | 2250 | 2.6 | 5 |

Remark: The elastic modulus value is called the code for the design of masonry building (GB50003-2011). Thus, an

isotropic elastic material is defined as follows: $E/G = 2(1 + \nu)$, where $\nu$ denotes the Poisson's ratio for $\nu = 0.3$, and

$E/G = 2.6$ (Burland, 1977).

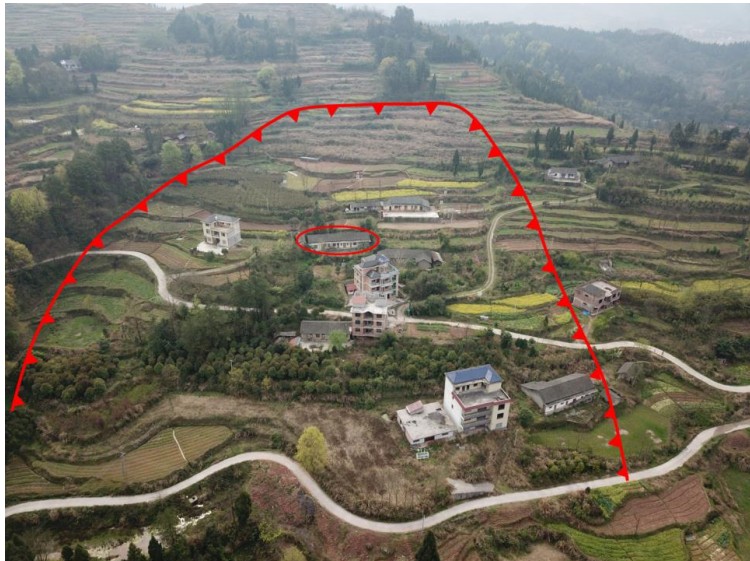

**Fig. 7. A typical example of a damaged building. The map is from Unmanned Air Vehicle operated by the author groups in the field.**




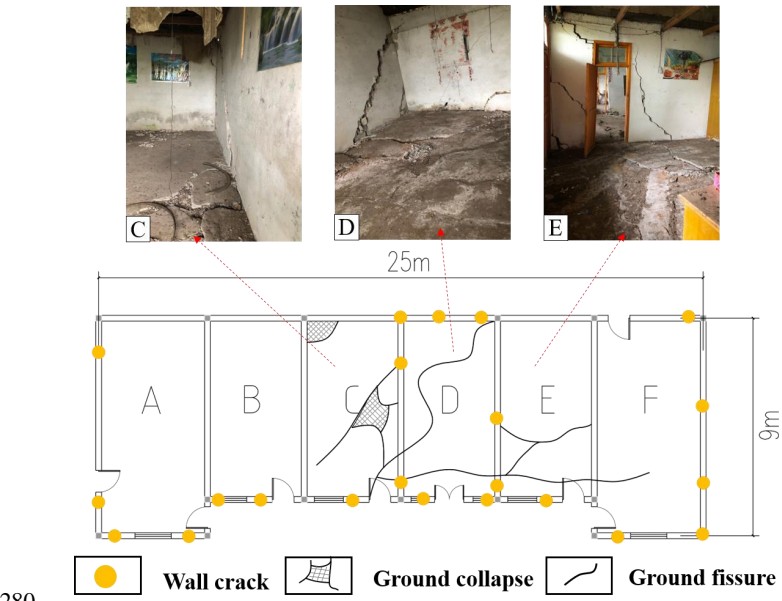


**Fig. 8. Floor plan of the object building.**

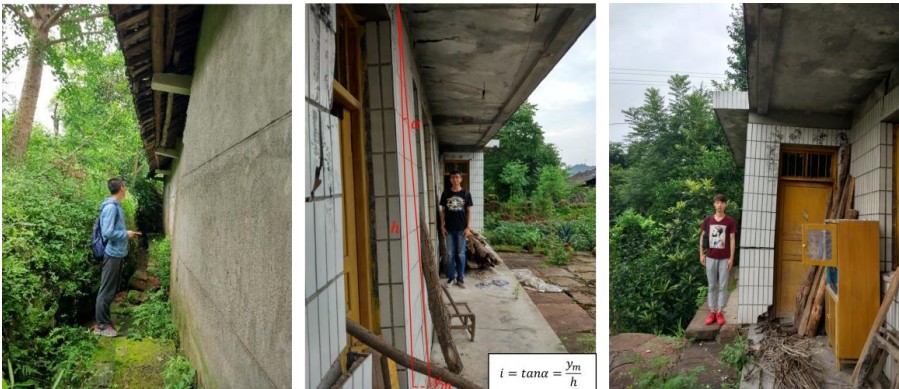

**Fig. 9. The integral decline state of the object building.**

**3.3 Rainfall data collection**

Landslides were usually induced by extreme or short-term sustained intense precipitation in this study area (Chen et al., 2014;

Qiong et al., 2018; Huang et al., 2014). Moreover, 3-day rainfall proved to be the most relevant parameter of landslide

occurrences in the study area (Lin et al., 2018 ). We collected the rainfall precipitation data of Sangzhi County from 1995 to

2016 (http://www.cma.gov.cn/). We applied the rainfall data for extreme rainfall analysis and scenario determination (Fig.

10).





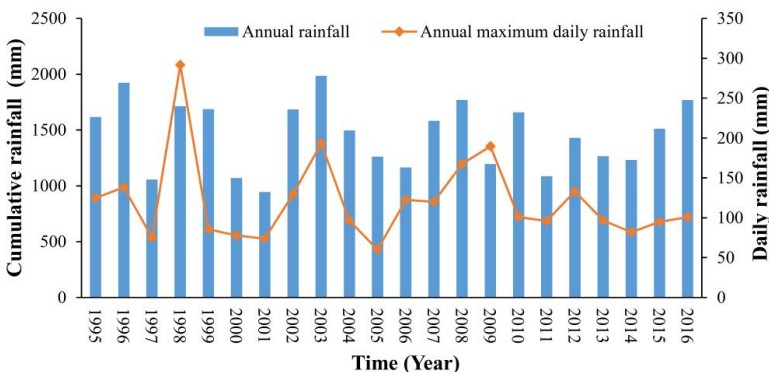


**Fig. 10. An average annual rainfall and maximum daily rainfall in the study area during the year of 1995–2016.**

## 4 Results

### 4.1 Extreme rainfall scenarios and landslide residual thrust calculation

The extreme rainfall distribution curve is depicted in Fig. 11 that is constructed by employing PT III distribution model and

the rainfall data. Using this curve, we can obtain the amount of 3-day cumulative precipitation corresponding to each return

period.

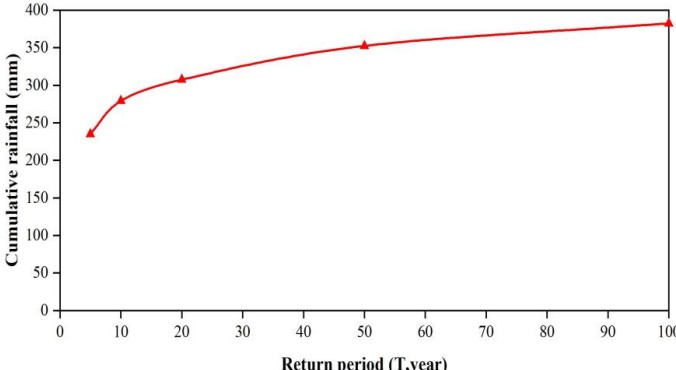

**Fig. 11. The extreme rainfall distribution curve.**

Groundwater levels based on three scenarios with different magnitudes of rainfall were selected: (a) dry condition

without earthquake; (b) rainfall with a return period of 5 years (3-day precipitation 235 mm from Fig. 11) without earthquake;

(c) rainfall with a return period of 10 years (3-day precipitation 279 mm from Fig. 11) without earthquake; (d) rainfall with a

return period of 50 years (3-day precipitation 352 mm from Fig. 11) without earthquakes. For scenarios b, c, and d, rainfall

precipitation was utilized as the boundary condition to simulate the groundwater level of the landslide.

The results of the residual thrust and the corresponding safety factor are presented in Table 5. These values were

obtained by the landslide residual force calculation method (section 2.1) for the geological profile (Fig. 5). In the dry season





(scenario a), the landslide performs residual thrust of 142 KN/m and safety factor of 0.853, while the three values significantly changed in rainy seasons (scenario b, c and d). For example, at least about fifteen times of the residual thrust and almost half decreases of the safety factor occurred in rainy season with a 50-year rainfall. This indicates an important influence of rainfall on landslide stability and the building's safety factor.

**Table 5. Landslide residual thrust, pushing force on the building's foundation, and damage degree of the building based on three scenarios ((a) dry condition without earthquake; (b) rainfall with a return period of 5 years (3-day precipitation 235 mm/d from Fig. 11); (c) rainfall with a return period of 10 years (3-day precipitation 279 mm/d from Fig. 11) without earthquake; (d) rainfall with a return period of 50 years (3-day precipitation 352 mm/d from Fig. 11) without earthquakes).**

| Scenarios | $F_s$ | F(KN/m) | $q$ ( KN/m ) | $i$ ( % ) | $V$ |
|-----------|-------|---------|--------------|-----------|-----|
| a | 0.853 | 142 | 28 | 0.053 | 0.053 |
| b | 0.529 | 1756 | 351 | 0.656 | 0.656 |
| c | 0.481 | 2040 | 408 | 0.762 | 0.762 |
| d | 0.428 | 2638 | 528 | 0.985 | 0.985 |

**4.2 Results of scenario-based vulnerability curve of the building**

As aforementioned in the landslide description (section 3.1) and demonstrated in the geological profile (Fig. 5), the sliding mass material is silty clay and block rocks. Therefore, the thrust distribution form can be approximately considered as rectangular based on Table 1. By applying the results of the horizontal component of landslide residual thrust (using the method in section 2.1) and the soil depth where the building is located (Table 4), the pushing force on the foundation can be calculated by the corresponding thrust distribution function.

Table 5 illustrates the results of pushing force on the foundation, inclination, and the building damage degree based on different scenarios. This indicates that the building's vulnerability is very low ($V = 0.053$) in the dry season, with a pushing force of 28 KN/m on the building's foundation. Additionally, in rainy seasons, the building experienced severe damage with the damage degree of 0.798 (10-year rainfall) or even 0.985 (50-year rainfall).

Using the four sets of data in Table 5, we constructed the physical vulnerability function, and the constants in Eq. 11

were determined by employing the Weibull function. The physical vulnerability of the target building on Manjiapo landslide is demonstrated in Fig. 12. We can observe that the physical vulnerability is very low when the landslide is stable with a safety factor greater than 1.0. Also, when the safety factor is lower than 1.0, the physical vulnerability rapidly increases, and it is approximately 1.0 when the reciprocal value of the safety factor is 2.5. By utilizing this curve, we can obtain the possible physical vulnerability of the building if the safety factor of the landslide is known. Therefore, we need to

demonstrate that the safety factor is for the local area where the target building is located, but not for the whole landslide as usual.
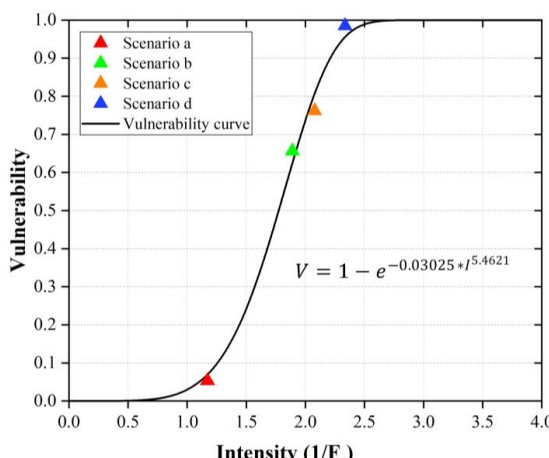

**Fig. 12. The physical vulnerability curve for masonry buildings impacted by the slow-moving landslides.**

**4.3 Influence of building characteristics on vulnerability and the sensitivity analysis**

To obtain the influence of factors on vulnerability, we conducted sensitivity analysis regarding building parameters. From

Table 4, we know that numerous parameters of the building were included in the building inclination and later damage

degree calculation: length, width, height, depth of foundation, and Young's modulus. The possible physical vulnerability of

the building with each changed parameter is depicted in Fig. 13.

As demonstrated in Fig. 13, we observe that the physical vulnerability is directly proportional to the building length and

is inversely proportional to the other parameters: building width and height, foundation depth and Young's modulus. This

indicates that the higher the ratio of building length and width, the more vulnerable to damage the building is. Besides,

buildings with more floors, deeper foundation, and higher Young's modulus can withstand less severe damage.

The results of the sensitivity analysis of the building parameters are demonstrated in Fig. 14. The red line that

represents length has the steepest slope among all the lines, indicating that the length of the building has the most significant

influence on the physical vulnerability of building. We can simultaneously obtain the second major factor that is the

foundation depth, while the third one is building width.

We tested four types of buildings with different lengths: 15 m, 20 m, 25 m, and 30 m (Fig. 15(a)). When FS is greater

than 1.0, the building physical vulnerability with any length is very low, that is, almost no damage. In addition, the building

demonstrated a different performance when FS is less than 1.0. The building physical vulnerability with length 15 m was

slightly increased when the landslide stability was getting worse. However, the building physical vulnerability with length 30

m rapidly increased when FS was less than 1.0. This indicates that the buildings on the location where the target building




stands have a limit length of 30 m. When the length of the building was greater than 30 m, the building faced severe damage

if FS was less than 1.0.

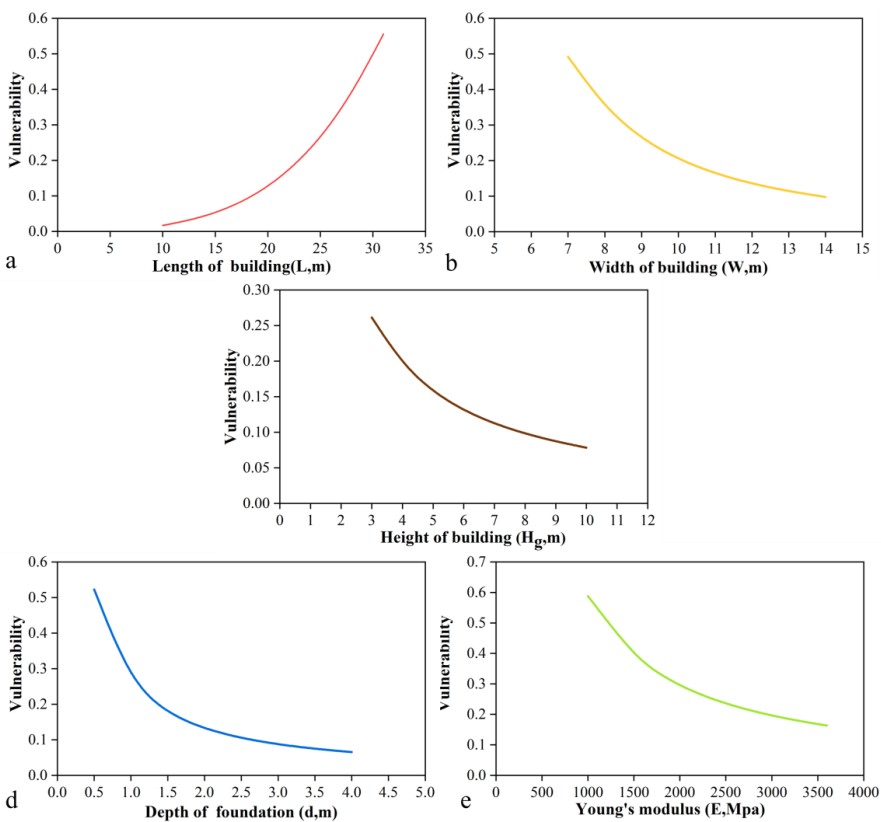

a b d e


**Fig. 13. Vulnerability curves for different building parameters: a) length, b) width, c) height, d) depth of foundation, and e) Young's modulus.**

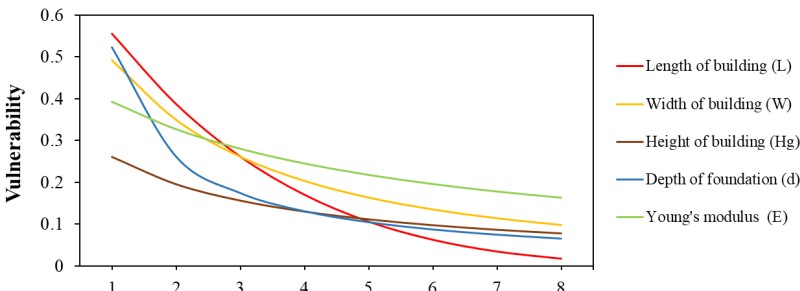

**Fig. 14. The sensitivity analysis of building parameters for physical vulnerability.**



To further test the detailed influences of the building parameters, we select the top two parameters based on the above results of sensitivity analysis: building length and foundation depth. Two sets of physical vulnerability curves are depicted in Fig. 15, and the corresponding functions of building physical vulnerability at the three scenarios are presented in Table 6.

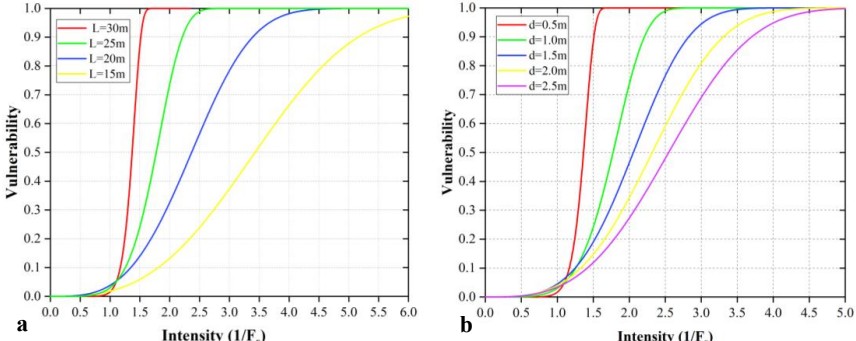

**Fig. 15. Physical vulnerability curves of buildings with different parameters: (a) length and (b) foundation depth.**

Physical vulnerability curves of buildings with various foundation depths are depicted in Fig. 15(b), while the physical vulnerability curves of buildings with various lengths are depicted in Fig. 15(a). The difference in the physical vulnerability of the buildings with different foundation depths is not significant when the FS is greater than 1.0. The tendency for the foundation depths over 1.0 m continues until the reciprocal value of FS is less than 1.25. Meanwhile, the building with foundation depth 1.0 is susceptible to the changes of FS. A rapid increase of building damage with such foundation depth

occurs when the FS is less than 1.0. This indicates a limit foundation depth for the location where the target building stands.

**Table 6. Physical vulnerability functions of buildings with different lengths and foundation depths based on various scenarios.**

| Parameters | Scenarios | $F_s$ | F (KN/m) | $i$ (%) | V | vulnerability function |
|---|---|---|---|---|---|---|
| Length (L/m) | 15 a | 0.853 | 142 | 0.010 | 0.010 | |
| | b | 0.529 | 1756 | 0.128 | 0.128 | $V = 1 - e^{-0.0184*(1/Fs)^{2.94457}}$ |
| | c | 0.481 | 2040 | 0.149 | 0.149 | |
| | d | 0.428 | 2638 | 0.193 | 0.193 | |
| | 20 a | 0.853 | 142 | 0.025 | 0.025 | |
| | b | 0.529 | 1756 | 0.312 | 0.312 | $V = 1 - e^{-0.03885*(1/Fs)^{3.34436}}$ |
| | c | 0.481 | 2040 | 0.362 | 0.362 | |
| | d | 0.428 | 2638 | 0.469 | 0.469 | |
| | 25 a | 0.853 | 142 | 0.053 | 0.053 | |
| | b | 0.529 | 1756 | 0.656 | 0.656 | $V = 1 - e^{-0.03025*(1/Fs)^{5.4621}}$ |
| | c | 0.481 | 2040 | 0.762 | 0.762 | |
| | d | 0.428 | 2638 | 0.985 | 0.985 | |
| | 30 a | 0.853 | 142 | 0.101 | 0.101 | |
| | b | 0.529 | 1756 | 1.239 | 1.000 | $V = 1 - e^{-0.01581*I(1/Fs)^{11.99869}}$ |
| | c | 0.481 | 2040 | 1.440 | 1.000 | |
| | d | 0.428 | 2638 | 1.862 | 1.000 | |
| Foundation depth (d/m) | 0.5 a | 0.853 | 142 | 0.106 | 0.106 | |
| | b | 0.529 | 1756 | 1.310 | 1.000 | $V = 1 - e^{-0.01662*(1/Fs)^{12.00525}}$ |
| | c | 0.481 | 2040 | 1.523 | 1.000 | |
| | d | 0.428 | 2638 | 1.969 | 1.000 | |
| | 1.0 a | 0.853 | 142 | 0.053 | 0.053 | $V = 1 - e^{-0.03025*(1/Fs)^{5.4621}}$ |



| | | | | | | |
|---|---|---|---|---|---|---|
| | b | 0.529 | 1756 | 0.656 | 0.656 | |
| | c | 0.481 | 2040 | 0.762 | 0.762 | |
| | d | 0.428 | 2638 | 0.985 | 0.985 | |
| 1.5 | a | 0.853 | 142 | 0.035 | 0.035 | $V = 1 - e^{-0.04566*(1/Fs)^{3.77289}}$ |
| | b | 0.529 | 1756 | 0.437 | 0.437 | |
| | c | 0.481 | 2040 | 0.508 | 0.508 | |
| | d | 0.428 | 2638 | 0.656 | 0.656 | |
| 2.0 | a | 0.853 | 142 | 0.027 | 0.027 | $V = 1 - e^{-0.04054*I^{3.37528}}$ |
| | b | 0.529 | 1756 | 0.328 | 0.328 | |
| | c | 0.481 | 2040 | 0.381 | 0.381 | |
| | d | 0.428 | 2638 | 0.492 | 0.492 | |
| 2.5 | a | 0.853 | 142 | 0.021 | 0.021 | $V = 1 - e^{-0.03439*I(1/Fs)^{3.21172}}$ |
| | b | 0.529 | 1756 | 0.262 | 0.262 | |
| | c | 0.481 | 2040 | 0.305 | 0.305 | |
| | d | 0.428 | 2638 | 0.394 | 0.394 | |

## 5 Discussion

We developed a scenario-based and mechanical method for analyzing the physical vulnerability of buildings with slow-moving landslides. The method enabled us to analyze the physical vulnerability from a mechanical view on soil-structure interaction, which can help us better understand the building damage on the slow-moving landslides.

The results of the application correspond to the fact from the field investigation: As was described in section 3.2, the building damage occurred from June 28th to 30th, 2016. The inclination measured in the field is from 0.7 (Fig. 9(a)) to 1.0 (Fig. 9(c)). During the rainy season, the daily precipitation was 82.4 mm/d (247 mm of 3-day precipitation) and has a return period of 10 years as depicted in the extreme rainfall curve (Fig. 11). Therefore, the calculated physical vulnerability is observed to be 0.762 in Table 5 that is close to the real building damage.

Herein, the influence of building parameters (length, width, height, foundation depth, etc.) on physical vulnerability corresponds to the other previously conducted studies (Li et al., 2010; Du et al., 2013; Corominas et al., 2014;). This is consistent with the study conducted by Corominas et al. (2014) that the typology of buildings is a key factor in physical vulnerability quantification. In this study, the physical vulnerability is inversely proportional to the building height. This agrees with the method proposed by Du (2013). In Du's study, the lower the height of the building, the more serious damage occurs on the building for a given depth of landslide. The same result can also be found in the study by Li et al. (2010).

Particularly, it was observed that the greater the building length, the more serious the building's damage is based on the same landslide force. When the length is close to 30 m, the building's damage is severe under unstable landslide (FS < 1.0). The target building length is 25 m in this study. The building damage was much, and it almost collapsed when the landslide occurred. Therefore, in the land-use planning for settlements on slopes, we suggest that it is important to choose the length–width ratio of buildings. We indicated that the building length perpendicular to the sliding direction of the landslide should not be too large. We note that 30 m is the threshold value for the length of masonry buildings in this case. Thus, the size



(length, width, height) of buildings should be designed in landslide-prone areas, which is also the way to control landslide risk in terms of decreasing physical vulnerability.

Furthermore, the shallower the foundation depth is, the more serious the building's damage when a given landslide is considered. The larger Young's modulus of the foundation is, the lower the building vulnerability of the building is. These results agree with the view of Corominas et al. (2014): that a deep foundation is less vulnerable than a shallow foundation; rigid foundations may be less vulnerable than flexible foundations. Moreover, the threshold value of the building's foundation was also obtained in this study. A 0.5 m is the threshold value in this case. Besides, the foundation depth depends

on the local geological conditions. The buildings with the slow-moving landslides require a deeper foundation depth, especially when the stratum is weak. Therefore, before houses should be built on a slope, both slope stability and lithology need to be considered.

  An interesting thing we should mention is that the procedure of the physical vulnerability estimation cannot only be used for the local scale landslide assessment but also can be used for the regional scale of landslide risk assessment. Since

the output of physical vulnerability is related to the safety factor of landslides of area where the building located, we can evaluate the physical vulnerability of buildings prone to slow-moving landslides at a regional scale. For instance, the distribution of FS can now be easily obtained by numerous researchers. Then, if we employ the physical vulnerability curves or the curves generated by the method of this study, the risk can be easily quantified for the potential losses of buildings.

  Some limitations of this study have to be mentioned here. First, in this study, the buildings analyzed are inside the

boundary of landslides. The physical vulnerability of buildings passing through landslide boundary is indicated to utilize displacement parameters as landslide intensity. The soil pressure on the foundation is suitable in this case. Second, the building's foundation is simplified as an integral and a simple beam. We did not consider the friction between the foundation and soil. More relative mechanical models are required in future studies. Finally, an uncertainty analysis was not conducted in this study. Random distribution of soil parameters for landslide FS calculation, such as shear strength, can be considered

for generating fragility curves based on this study.

## 6 Conclusions

This study proposed a method for constructing physical vulnerability curves and functions by utilizing the analysis of the horizontal force of landslide acting on the foundation and the physical response of the building. The proposed method was applied to slow-moving landslides in China, in which a severely damaged building was considered as the target structure and

measured in the field.
The proposed method mainly comprises of calculating the landslide safety factor and horizontal load on foundations based on different scenarios (extreme rainfall with different return periods in this study); we then analyzed the physical response of foundation and the integral inclination of building. Finally, we generated the physical vulnerability curves and constructed functions by applying the Weibull function.

Good consistency between the estimated building physical vulnerability and on-field damage evidence was observed on the target case building. In particular, by the sensitivity analysis conducted, the main two contributions of building characteristics for the physical vulnerability were observed to be the building length and foundation depth. Two sets of physical vulnerability, between the damage degree and landslide safety factor for the area where the buildings are located, were separately generated by considering the two parameters. We hope that this study can be a useful supplement for the

physical vulnerability estimation of buildings in the area prone to slow-moving landslides.

***Data availability.*** The study relied on two sets of data: (i) the data collected by the field work, (ii) the detailed landslide investigation reports provided by the China Geological Survey. The data is included in Section 3 in this paper. The relevant datasets in this study are available from the corresponding author on reasonable request.

***Author contribution.*** Qin Chen and Lixia Chen discussed the research plan, carried out the field work, taken the modelling

and wrote the paper. Qin Chen prepared the figures of the paper. Lixia Chen and Kunlong Yin supervised the research. Lei Gui and Xuelian Cao helped in modelling. Lei Gui and Juan Du helped in data collection. Shrestha helped in the paper development and English writing.

***Competing interests.*** The authors declare that they have no conflict of interest.

***Acknowledgements.*** This research is partially supported by the project "Studies on spatial-temporal differences of large

accumulation landslide deformation and its vulnerability model for buildings in the Three Gorges reservoir" which is financed by the National Natural Science Foundation of China [Grant No. 41877525], and partially supported by the project "Study on the dynamic response of the quantitative vulnerability of buildings in different evolution stage of landslides" which is financed by the National Natural Science Foundation of China [Grant No. 41601563].

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
