# Peer review of "Assessment of the physical vulnerability of buildings affected by slow-moving landslides"

_Natural Hazards and Earth System Sciences, 2019_

## Referee Comment (RC1) · Anonymous Referee #1 · 28 Feb 2020

General comments

The paper shows the results of a study focused on the vulnerability analysis of buildings exposed to slow-moving landslides. To this aim, a methodological approach – which involves carrying out numerical analyses based on different rainfall scenarios – is proposed and applied to a case study in China. The addressed topic is significant. However, much effort should be done in clarifying and deepening some addressed issues in order to allow for the exportability of obtained results.

Specific comments

In Section 2.2.1 the Authors recall the equivalent elastic beam – originally introduced by Burland and Wroth (1974) to define a damageability criterion – in order to compute the

maximum deflection exhibited by the same beam under a uniform load whose modulus equals q. In Figure 2 this uniform load acts horizontally, in correspondence of the lateral surface of the building's foundation affected by the landslide; whereas in Figure 3 the uniform load is applied vertically to the elastic beam. Accordingly, it is not clear in which direction the maximum deflection develops. Furthermore, symbols adopted in Figures 2 and 3 to denote the geometrical characteristics of the building's foundation are not internally consistent. Could the Authors better explain?

In the same Section 2.2.1 the concept of "inclination" of a building is introduced. Does this inclination corresponds to the "rotation" or "slope" (i.e. the change in gradient of a line joining two reference points of the foundation base) or to the "tilt" (describing the rigid body rotation of the whole superstructure or a well-defined part of it) defined by Burland and Wroth (1974)? Or does it refer to another well-defined parameter? Please clarify. I also suggest to associate the Eq. (9) – used to express mathematically the concept of inclination – with a Figure helpful to better understand the meaning of symbols adopted in Eq. (9), including the angle alpha.

In Table 3 the shear strength parameters of soils involved in the shear zone of the Manjiapo landslide are summarized. Are they residual shear strength values? And, more in general, what type of laboratory tests was carried out? Please explain.

In Table 4 it is not clear if the Young's modulus refers to the masonry constituting the building superstructure or to the material constituting the building foundation. Please clarify.

In Section 4.1 the rainfall scenarios considered for transient seepage analyses are introduced. However, relevant information is provided neither on the fixed boundary conditions nor on the adopted hydraulic conductivities. Please improve this Section.

In Table 5 the results obtained for the four considered rainfall scenarios are summarized. In all the cases, the factor of safety (FS) is lower than 1. This would imply that the landslide is always moving, in disagreement with the information gathered by the

Authors about the cracks on the ground surface of the Manjiapo landslide. In particular, the Authors observe that "Meanwhile, since the extreme rainfall events were recently rare, the deformation of the landslide did not obviously change, which was similar to the deformation situation in June 2016. For example, the cracks on the landslide did not expand, and the number of new cracks was very few" (page 10 – lines from 249 to 252). Probably, the shear strength parameters used for the limit equilibrium analyses are too low (see Table 3) and should be compared with those deriving from the back-analysis of the event occurred on June 2016.

In Section 4.2 the results of the vulnerability analysis concerning a selected building within the affected area of the Manjiapo landslide are presented. Focusing on the obtained vulnerability curve (Fig. 12) the Authors observe that "the physical vulnerability is very low when the landslide is stable with a safety factor greater than 1.0" (lines from 326 to 327). How is this observation justifiable? Indeed, it is expected that the building vulnerability equals 0 (no damage) if the landslide does not move. In this regard, are the Authors sure that the chosen Weibull (1951) function is the best one to mathematically express the vulnerability curve when 1/FS is adopted as landslide intensity parameter?

In the Discussion, the Authors stress that "the physical vulnerability is inversely proportional to the building height" (line 384). This is not in agreement with thresholds values of the building inclination summarized in Table 2. Indeed, as the building height (from the outdoor ground) increases the threshold value decreases.

In my opinion, the vulnerability curves shown in Figure 15 have to be further validated before applying them in analyses at regional scale (lines from 403 to 408).

Technical corrections

The symbol adopted throughout the manuscript to indicate the unit of measurement of force should be "kN" (lowercase k) instead of "KN".

The symbol adopted in Table 3 to indicate the unit of measurement of stress should be "kPa" instead of "kpa".

In the Note of Table 3 it is not clear if the information provided about the "permeability coefficient" and the "volume of water content" (or "volumetric water content"?) refers to the considered soil in saturated conditions or not. Relevant units of measurement should be provided, if applicable.
* * *

---

## Referee Comment (RC2) · Anonymous Referee #2 · 2 Mar 2020

General comments The paper presents a method for assessing the physical vulnerability of buildings located in slow-moving landslide-affected areas. The Authors develop multi-scenario analyses to generate physical vulnerability curves and they validate their results on a building impacted by a slow-moving landslide in a study area in China. The work is completed by a sensitivity analysis on the parameters playing a major role on the vulnerability of the analyzed structure. The addressed topic is timely and of interest to the landslide community for its potential applicability. However, the current version of the paper needs to be thoroughly revised to better clarify some of the assumptions made and the definition of parameters according to the literature. Since the Authors state that the procedure could be easily exported to other landslides even over large areas, an effort must be made to allow the reader catch the (minimum number) of input

parameters that are necessary to perform similar studies. Some English sentences are unclear; accordingly, a review from a native speaker would be helpful.

Specific Comments As for specific and technical comments, the Authors can refer to the annotated pdf file in attachment. Overall, some key issue are synthesized in the following: - the Abstract should be rearranged and totally rewritten. It cannot be a list of steps followed during the analysis. Authors should mention the problem and the approach followed to get the results. - Fig.3, pag.6. Authors should better clarify, for instance with an additional Figure, how the lateral forces impacting the foundation can be associated with ym (that is the inflection under vertical loads). At the moment the concept of im is not clear. - The sentence (pag.7) referring to Finno et al. (2005) should be better clarified. - The Authors identify the damage classification with vulnerability. This aspect deserves further clarifications based on widely shared literature. - Provide more details on laboratory tests used to gather the values of the shear strength parameters shown in Table 3. - In Figure 12, it is not clear the range of variation of 1/Fs. Some further comments would be helpful. - The comparison shown in Figure 13 are unclear. What is there on x-axis? - Conclusions: please clarify better or add references concerning the calculation of FS just in correspondence of buildings and over large areas. - Exportability should be better supported with clarifications.

Please also note the supplement to this comment:
https://www.nat-hazards-earth-syst-sci-discuss.net/nhess-2019-318/nhess-2019-318-RC2-supplement.pdf

**Supplement:**

[revised manuscript text omitted]

---

## Author Comment (AC1) · 23 Mar 2020

Dear Referee, We would like to thank you for your professional and constructive comments concerning our manuscript entitled "Assessment of the physical vulnerability of buildings affected by slow-moving landslides". These comments are all valuable and helpful for revising and improving our manuscript. We have seriously considered and provided our point-by-point responses, which are listed below.

Specific comments: (1) In Section 2.2.1 the Authors recall the equivalent elastic beam – originally introduced by Burland and Wroth (1974) to define a damageability criterion – in order to compute the maximum deflection exhibited by the same beam under a uniform load whose modulus equals q. In Figure 2 this uniform load acts horizontally, in correspondence of the lateral surface of the building's foundation affected by the landslide; whereas in Figure 3 the uniform load is applied vertically to the elastic beam. Accordingly, it is not clear in which direction the maximum deflection develops. Furthermore, symbols adopted in Figures 2 and 3 to denote the geometrical characteristics of the building's foundation are not internally consistent. Could the Authors better explain?

Response: thank you very much for the comment. In Figure 3, we actually try to express the uniform load applied horizontally, so sorry for the confusing. We plan to revise the figure 3 as follows.

Fig.3 The simple beam with its foundation affected by landslide thrust. Fig. 2. Schematic diagram of landslide thrust action on a building. where q denotes the distribution force on the foundation (kN/m), F denotes the horizontal component of landslide residual thrust (Pi) in Eq. (3), and h denotes the vertical distance from sliding surface to the ground surface. i denotes the inclination of the building, which is the ratio of the maximum horizontal deformation ym to the height Hg of the building calculated from the outdoor ground (Fig.2). L, W, and d denote the length, width and depth of the building foundation (Fig.3).

(2) In the same Section 2.2.1 the concept of "inclination" of a building is introduced. Does this inclination corresponds to the "rotation" or "slope" (i.e. the change in gradient of a line joining two reference points of the foundation base) or to the "tilt" (describing the rigid body rotation of the whole superstructure or a well-defined part of it) defined by Burland and Wroth (1974)? Or does it refer to another well-defined parameter? Please clarify. I also suggest to associate the Eq. (9) – used to express mathematically the concept of inclination – with a Figure helpful to better understand the meaning of symbols adopted in Eq. (9), including the angle alpha.

Response: thank you very much for the suggestion. The concept of "inclination" in our manuscript is that the ratio of the difference ym (between the top and bottom of the

building) to the height Hg of the building. In some references, such as researches by Li (2010), this concept is used for "inclination". We inserted a figure in Fig.2 to better understand the meaning of symbols adopted in Eq. (9) as follow.

Reference 1. Li, Z., Nadim, F., Huang, H., Uzielli, M., and Lacasse, S.: Quantitative vulnerability estimation for scenario-based landslide hazards, Landslides, doi:10.1007/s10346-009-0190-3, 2010

Fig. 2. Schematic diagram of landslide thrust action on a building.

(3) In Table 3 the shear strength parameters of soils involved in the shear zone of the Manjiapo landslide are summarized. Are they residual shear strength values? And, more in general, what type of laboratory tests was carried out? Please explain.

Response: thank you very much for the comment. The shear strength parameters in Table 3 are residual values. According to the report provided by the China Geological Survey (Hunan Institute of Xiangxi Geological Engineering Survey) in 2017, six groups of undisturbed soil samples were collected from the shear zone of the Manjiapo Landslide. Obtained by residual shear tests in the laboratory, the shear strength parameters of slip soils in Table 3 are the average values of these six groups of soil samples.

(4) In Table 4 it is not clear if the Young's modulus refers to the masonry constituting the building superstructure or to the material constituting the building foundation. Please clarify.

Responses: Thank you very much for your comments. The Young's modulus in Table 4 refers to the material constituting the building foundation. We will revise Table 4 as follows.

Table 4. Parameters of the case building and its foundation on the Manjiapo landslide For building For foundation Soil depth where the building located (m) Length L (m) Width W (m) Height Hg (m) Depth d (m) Young's modulus E (MPa) Shear modulus G (MPa) E/G 25 9 2.8 1 2250 865 2.6 5

(5) In Section 4.1 the rainfall scenarios considered for transient seepage analyses are introduced. However, relevant information is provided neither on the fixed boundary conditions nor on the adopted hydraulic conductivities. Please improve this Section.

Response: thank you very much for the suggestion. We use the SEEP/W code (GEOSTUDIO) to analyze the groundwater seepage of Manjiapo landslide. We obtain the amount of 3-day cumulative precipitation corresponding to each return period by using PT (Pearson type) âĚć distribution model and the rainfall data (Fig. 11). The average amount of 3-day cumulative precipitation is input to the software in turn, and then the groundwater under the rainfall scenarios is simulated.

The saturated volumetric water content is 0.4 by cutting ring method. The saturated permeability coefficient is obtained by back analysis. We choose the saturated volumetric water content and the permeability coefficient by the variable-controlling approach. Three groups of input values are: 0.4, 0.1; 0.4, 0.2; 0.4, 0.3. Then, the groundwater is simulated and then validated for the rainfall event in March 2018. The root mean square error (RMSE) is utilized to check the accuracy of calculation. Lower RMSE means smaller error and better prediction effect. The results of RMSE are shown in the following table. We find the saturated volumetric water content is 0.4 and the most suitable value of permeability coefficient is 0.3 m/d.

Table Permeability coefficient back analysis of the rainfall event in March 2018, by comparing the root mean square errors (RMSE) of for three hydrological gauges (installed by the authors in December 2017, see Fig.5) on Manjiapo landslide The permeability coefficient (m/d) 0.1 0.2 0.3 RMSE (STK-1) 2.280 2.222 2.154 RMSE (STK-2) 0.860 0.677 0.615 RMSE (STK-3) 2.540 2.491 2.405 Note: the saturated volumetric water content by Lab test is 0.4.

Fig. 5. Geological profile of âĚă-âĚă' of the Manjiapo landslide (1:1 000). (6) In Table 5 the results obtained for the four considered rainfall scenarios are summarized. In all the cases, the factor of safety (Fs) is lower than 1. This would imply that the landslide is always moving, in disagreement with the information gathered by the Authors about the cracks on the ground surface of the Manjiapo landslide. In particular, the Authors observe that "Meanwhile, since the extreme rainfall events were recently rare, the deformation of the landslide did not obviously change, which was similar to the deformation situation in June 2016. For example, the cracks on the landslide did not expand, and the number of new cracks was very few" (page 10 – lines from 249 to 252). Probably, the shear strength parameters used for the limit equilibrium analyses are too low (see Table 3) and should be compared with those deriving from the back-analysis of the event occurred on June 2016.

Response: thank you very much for the suggestion. The factor of safety (Fs) in the original Table 5 is for the area where the target building is located, but not for the whole landslide. This is pointed out by the sentence from line 184 to 185 on page 8. To avoid confusing, we revised this variable to be Fsb. In addition, we add the results of safety factors for the whole landslide under four scenarios in table 5 as follows. Now in this table, Fs refers to the factor safety of the whole landslide. From table 5, we find that the landslide is stable with factor of safety 1.457.

Table 5. Landslide residual thrust, pushing force on the building's foundation, and damage degree of the building based on four scenarios: (a) dry condition without earthquake, (b) rainfall with a return period of 5 years (3-day precipitation 235 mm/d from Fig. 11), (c) rainfall with a return period of 10 years (3-day precipitation 279 mm/d from Fig. 11) without earthquake, and (d) rainfall with a return period of 50 years (3-day precipitation 352 mm/d from Fig. 11) without earthquakes. Scenarios Fsb Fs F(KN/m) qïijĹKN/mïijĽ ïïijĹ%ïijĽ V a 0.853 1.457 142 28 0.053 0.053 b 0.529 0.819 1756 351 0.656 0.656 c 0.481 0.778 2040 408 0.762 0.762 d 0.428 0.632 2638 528 0.985 0.985

Note: Fsb is the safety factor of the area where the target building is located.

(7) In Section 4.2 the results of the vulnerability analysis concerning a selected building within the affected area of the Manjiapo landslide are presented. Focusing on the obtained vulnerability curve (Fig. 12) the Authors observe that "the physical vulnerability is very low when the landslide is stable with a safety factor greater than 1.0" (lines from 326 to 327). How is this observation justifiable? Indeed, it is expected that the building vulnerability equals 0 (no damage) if the landslide does not move. In this regard, are the Authors sure that the chosen Weibull (1951) function is the best one to mathematically express the vulnerability curve when 1/FS is adopted as landslide intensity parameter?

Response: thank you very much for the comment. For slow-moving landslides, they can have a Fs greater than 1.0 but with cracks within the landslide area. According to the standard of Code for geological investigation of landslide prevention (GB/T32864ï¡đ2016) , when the whole landslide has Fs value from 1.0 to 1.05, the landslide will have small scale deformation or cracks. While the buildings located across the cracks can have damages with a certain degree. In the Three Gorges Reservoir area, China (Chen et, 2016 ) and other areas, such as Moio della Civitella (Salerno province, Italy) (Infante et al., 2016), the buildings on the huge, slow-moving landslides will appear this state. So, to solve the problem on building's vulnerability, we need to focus on the local stability of this kind of landslide like Manjiapo landslide, but not the whole body. In Figure 12, the Fs value is for the local stability of the soil where the case building located. Following the above question, we need to modify Fs in Figure 12 to be Fsb. We can find from Figure 12 and Figure 15 that, when 1.0 < Fsb < 1.05, the building vulnerability is from 0 to 0.1. This means the building is damaged very slightly, which is consistent with the real state of buildings on slow-moving landslides.

In this regard, we are sure that Weibull function is suitable to express the vulnerability curve. In fact, Weibull function is used to express the vulnerability curve in many present literatures, such as Dario Peduto et (2017), Kang et (2016), Papathoma-Köhle (2016), Negulescu et (2010).

Reference 1. Chen, L., Cao, X., Yin, K., Wu, Y., and Li, Y.: Physical vulnerability assessment for buildings impacted by a slow moving landslide based on field work

and statistical modelling, in: Landslides and Engineered Slopes. Experience, Theory and Practice, 2016. 2. Infante, D., Confuorto, P., Di Martire, D., Ramondini, M. and Calcaterra, D.: Use of DInSAR Data for Multi-level Vulnerability Assessment of Urban Settings Affected by Slow-moving and Intermittent Landslides, Procedia Engineering, 158, 470–475, doi:10.1016/j.proeng.2016.08.474, 2016. 3. Peduto, D., Ferlisi, S., Nicodemo, G., Reale, D., Pisciotta, G., and Gullà, G.: Empirical fragility and vulnerability curves for buildings exposed to slow-moving landslides at medium and large scales, Landslides, doi:10.1007/s10346-017-0826-7, 2017. 4. Kang, H. sub and Kim, Y. tae: The physical vulnerability of different types of building structure to debris flow events, Natural Hazards, 80(3), 1475–1493, doi:10.1007/s11069-015-2032-z, 2016. 5. Papathoma-Köhle, M.: Vulnerability curves vs. Vulnerability indicators: Application of an indicator-based methodology for debris-flow hazards, Natural Hazards and Earth System Sciences, doi:10.5194/nhess-16-1771-2016, 2016. 6. Negulescu, C. and Foerster, E.: Parametric studies and quantitative assessment of the vulnerability of a RC frame building exposed to differential settlements, Nat. Hazards Earth Syst. Sci., 10(9), 1781–1792, 2010.

(8) In the Discussion, the Authors stress that "the physical vulnerability is inversely proportional to the building height" (line 384). This is not in agreement with thresholds values of the building inclination summarized in Table 2. Indeed, as the building height (from the outdoor ground) increases the threshold value decreases.

Response: thank you very much for the comment. Table 2 expresses threshold values of the building inclination for three types of buildings with different height. In this manuscript, we focus on the first class of building with height lower than 24 m, which are common residential buildings in rural areas of China. As to this kind of building, the threshold is a fixed value 1% and the physical vulnerability is inversely proportional to the building height according to the result form Figure 13c.

Table 2. The threshold value of building inclination (Ministry of Housing and Urban–Rural Development of PRC, 2016).

Height HgïijĹmïijĽ H_g≤24 24<H_g≤60 60<H_g≤100 Threshold value i_m 1% 0.7% 0.5% (9) In my opinion, the vulnerability curves shown in Figure 15 have to be further validated before applying them in analyses at regional scale (lines from 403 to 408).

Response: thank you very much for the good comment. We are currently doing the researches on regional scale slow-moving landslide risk assessment in the Three Gorges reservoir area, China, which involves regional scale vulnerability assessment for buildings. We totally agree with you that before applying the results from this manuscript, we will do further validation.

We tried our best to improve the manuscript and made changes in the manuscript. We feel great thanks for your professional review work on our article, and hope that the responses will meet with approval.

Sincerely, Lixia Chen

Please also note the supplement to this comment:
https://www.nat-hazards-earth-syst-sci-discuss.net/nhess-2019-318/nhess-2019-318-AC1-supplement.pdf

—————————————

[Figure]

Fig. 2. Schematic diagram of landslide thrust action on a building.

[Figure]

Fig.3 The simple beam with its foundation affected by landslide thrust.

where $q$ denotes the distribution force on the foundation (kN/m), $F$ denotes the horizontal component of landslide residual thrust ($P_t$) in Eq. (3), and $h$ denotes the vertical distance from sliding surface to the ground surface. $i$ denotes the inclination of the building, which is the ratio of the maximum horizontal deformation $y_m$ to the height $H_g$ of the building calculated from the outdoor ground (Fig.2). $L$, $W$, and $d$ denote the length, width and depth of the building foundation (Fig.3).

Table 4. Parameters of the case building and its foundation on the Manjiapo landslide

| For building | | | | For foundation | | | Soil depth |
|---|---|---|---|---|---|---|---|
| Length | Width | Height | Depth | Young's modulus | Shear modulus | $E/G$ | where the building |
| $L$ (m) | $W$ (m) | $H_g$ (m) | $d$ (m) | $E$ (MPa) | $G$ (MPa) | | located (m) |
| 25 | 9 | 2.8 | 1 | 2250 | 865 | 2.6 | 5 |

**Fig. 1.**

**Table Permeability coefficient back analysis of the rainfall event in March 2018, by comparing the root mean square errors (RMSE) of for three hydrological gauges (installed by the authors in December 2017, see Fig.5) on Manjiapo landslide**

| The permeability coefficient (m/d) | 0.1 | 0.2 | 0.3 |
|---|---|---|---|
| RMSE (STK-1) | 2.280 | 2.222 | 2.154 |
| RMSE (STK-2) | 0.860 | 0.677 | 0.615 |
| RMSE (STK-3) | 2.540 | 2.491 | 2.405 |

Note: the saturated volumetric water content by Lab test is 0.4.

[Figure]

Fig. 5. Geological profile of I-I' of the Manjiapo landslide (1:1 000).

**Table 5. Landslide residual thrust, pushing force on the building's foundation, and damage degree of the building based on four scenarios: (a) dry condition without earthquake, (b) rainfall with a return period of 5 years (3-day precipitation 235 mm/d from Fig. 11), (c) rainfall with a return period of 10 years (3-day precipitation 279 mm/d from Fig. 11) without earthquake, and (d) rainfall with a return period of 50 years (3-day precipitation 352 mm/d from Fig. 11) without earthquakes.**

| Scenarios | $F_{sb}$ | $F_s$ | F(KN/m) | $q$(KN/m) | $i$(%) | $V$ |
|---|---|---|---|---|---|---|
| a | 0.853 | 1.457 | 142 | 28 | 0.053 | 0.053 |
| b | 0.529 | 0.819 | 1756 | 351 | 0.656 | 0.656 |
| c | 0.481 | 0.778 | 2040 | 408 | 0.762 | 0.762 |
| d | 0.428 | 0.632 | 2638 | 528 | 0.985 | 0.985 |

Note: $F_{sb}$ is the safety factor of the area where the target building is located.

**Table 2. The threshold value of building inclination (Ministry of Housing and Urban–Rural Development of PRC, 2016).**

| Height $H_g$(m) | $H_g \leq 24$ | $24 < H_g \leq 60$ | $60 < H_g \leq 100$ |
|---|---|---|---|
| Threshold value $i_m$ | 1% | 0.7% | 0.5% |

**Fig. 2.**

---

## Author Comment (AC2) · 23 Mar 2020

Dear Referee, We would like to thank you for your professional and constructive comments concerning our manuscript entitled "Assessment of the physical vulnerability of buildings affected by slow-moving landslides". These comments are all valuable and helpful for revising and improving our manuscript. We have seriously considered and provided our point-by-point responses, which are listed below.

(1) The abstract should be rearranged and totally rewritten. It cannot be a list of steps followed during the analysis. Authors should mention the problem and the approach followed to get the results.

Response: Thank you for your good comments. We will rearrange and rewritten the

abstract, which will mention the problem and the approach followed to get the results.

(2) Fig.3, pag.6. Authors should better clarify, for instance with an additional Figure, how the lateral forces impacting the foundation can be associated with ym (that is the inflection under vertical loads). At the moment the concept of im is not clear.

Response: Thank you for your good comments. Figure 3 on page 6 did not clearly express the direction of lateral forces impacting the foundation. We try to express the uniform load applied horizontally, so sorry for the confusing. We will modify this figure as follows.

The concept of im is the threshold value of inclination of buildings. Buildings with inclination exceeding im are dangerous and uninhabitable. In Table 2, we listed out the standard of threshold values for three kinds of buildings with different height. We have supplemented the figure in Fig.2 (see below) to illustrate the concept more clearly.

Fig. 2. Schematic diagram of landslide thrust action on a building.

Fig.3 The simple beam with its foundation affected by landslide thrust.

(3)The sentence (pag.7) referring to Finno et al. (2005) should be better clarified.

Response: Thank you for your good comments. The sentence (Page 7) is: Since cracks on walls are not visible, especially when the building with high stiffness is exceedingly inclined because of the ground deformation, they usually serve as the indicators of damage degree evaluation if the building stiffness is small (Finno et al., 2005)

Sorry for the confused expression in the above sentence. We want to clarify that cracks on building walls are not the only indicator to assess damage degree or vulnerability, especially when the building has a very good stiffness. So, we revised this sentence as follows.

Finno et al. (2005) found that when the buildings with high stiffness are seriously inclined due to the ground deformation, the wall cracking phenomenon is not obvious; On the contrary, if the stiffness of the building is small, the wall cracks seriously. This research indicated that if we only use cracks as indicator for vulnerability assessment, it is unsuitable. Other indicators, such as inclination, should be also taken into consideration.

(4) The Authors identify the damage classification with vulnerability. This aspect deserves further clarifications based on widely shared literature.

Response: Thank you for your good comments. In order to simplify the research work, many researchers directly use damage degree as vulnerability. Tarbotton et al. (2015) defined empirical vulnerability functions as "a continuous curve associating the intensity of the hazard (X-axis) to the damage response of a building (Y-axis)". Kang et al. (2016) think that the range of damage to the buildings makes it possible to assess the vulnerability using a vulnerability curve that relates the intensity of debris flow with the degree of damage. They use the degree of damage to the buildings to estimate vulnerability.

Reference 1. Tarbotton, C., Dall'osso, F., Dominey-Howes, D., Goff, J. The use of empirical vulnerability functions to assess the response of buildings to tsunami impact: comparative review and summary of best practice. Earth Sci. Rev. 142, 120–134, doi.org/10.1016/j.earscirev.2015.01.002, 2015. 2. Kang, H. sub and Kim, Y. tae: The physical vulnerability of different types of building structure to debris flow events, Natural Hazards, 80(3), 1475–1493, doi:10.1007/s11069-015-2032-z, 2016.

(5) Provide more details on laboratory tests used to gather the values of the shear strength parameters shown in Table 3.

Response: Thank you for your good comments. The shear strength parameters in Table 3 are residual values. According to the report provided by the China Geological Survey (Hunan Institute of Xiangxi Geological Engineering Survey) in 2017, six groups of undisturbed soil samples were collected from the shear zone of the Manjiapo Landslide. Obtained by residual shear tests in the laboratory, the shear strength parameters of slip soils in Table 3 are the average values of these six groups of soil samples.

(6) In Figure 12, it is not clear the range of variation of 1/Fs. Some further comments would be helpful.

Response: thank you very much for the suggestion. Based on the Chinese standard of Code for geological investigation of landslide prevention (GB/T32864ï¡đ2016), the landslide stability state can be classified into three according to the safety factor (Fs) of landslide. Please see more details in the following table.

(7) The comparison shown in Figure 13 is unclear. What is there on x-axis?

Response: thank you very much for the comment. We think you want to comment on Figure 13, which is used to compare the sensitivity of building characteristics on vulnerability. In this figure, the x-axis expresses sample's No. but not means real value. By putting the five parameters together on a single diagram, we can clearly compare and find out which parameter is more sensitive to vulnerability.

(8) Conclusions: please clarify better or add references concerning the calculation of FS just in correspondence of buildings and over large areas. - Exportability should be better supported with clarifications.

Response: thank you very much for the suggestion. We are currently doing the researches on regional scale slow-moving landslide risk assessment in the Three Gorges reservoir area, China, which involves regional scale vulnerability assessment for buildings. The topic in this manuscript is partially new. There are rare references presently concerning Fs of slow-moving landslides and vulnerability of buildings. But the researches about calculation of Fs over large areas can be found from some researches, such as Muntohar AS, Liao HJ (2009), Apip, Takara K, Yamashiki Y, et al (2010), Salciarini (2006) and Sorbino (2010). We are eager to link the intensity of slow-moving landslides with vulnerability of buildings over large areas. Before applying the results

from this manuscript, we will do further validation.

Reference 1. Muntohar AS, Liao HJ.: Analysis of rainfall-induced infinite slope failure during typhoon using a hydrological-geotechnical model. Environ Geol 56:1145–1159, 2009 2. Apip, Takara K, Yamashiki Y, et al.: A distributed hydrological-geotechnical model using satellite-derived rainfall estimates for shallow landslide prediction system at a catchment scale. Landslides 7:237–258, 2010 3. Salciarini, D., Godt, J. W., Savage, W. Z., Conversini, P., Baum, R. L. and Michael, J. A.: Modeling regional initiation of rainfall-induced shallow landslides in the eastern Umbria Region of central Italy, Landslides, doi:10.1007/s10346-006-0037-0, 2006. 4. Sorbino, G., Sica, C. and Cascini, L.: Susceptibility analysis of shallow landslides source areas using physically based models, Natural Hazards, doi:10.1007/s11069-0, 2010

We tried our best to improve the manuscript and made changes in the manuscript. We feel great thanks for your professional review work on our article, and hope that the responses will meet with approval.

Sincerely, Lixia Chen

Please also note the supplement to this comment:
https://www.nat-hazards-earth-syst-sci-discuss.net/nhess-2019-318/nhess-2019-318-AC2-supplement.pdf
* * *
[Figure]

**Fig. 2. Schematic diagram of landslide thrust action on a building.**

**Fig.3 The simple beam with its foundation affected by landslide thrust.**

where $q$ denotes the distribution force on the foundation (kN/m), $F$ denotes the horizontal component of landslide residual thrust ($P_z$) in Eq. (3), and $h$ denotes the vertical distance from sliding surface to the ground surface. $i$ denotes the inclination of the building, which is the ratio of the maximum horizontal deformation $y_m$ to the height $H_g$ of the building calculated from the outdoor ground (Fig.2). $L$, $W$, and $d$ denote the length, width and depth of the building foundation (Fig.3).

**Fig. 1.**

Table . The range of safety factor ($F_s$) of landslide and its state

| The safety factor $F_s$ | $0 <F_s< 1.00$ | $1.00 \leq F_s < 1.05$ | $F_s \geq 1.05$ |
|---|---|---|---|
| $1/F_s$ | $1/F_s > 1.00$ | $0.95 < 1/F_s \leq 1.00$ | $1/Fs \leqslant 0.95$ |
| Stability state of landslide | unstable | Less stable | stable |
| Description | (1) Many newly expanded cracks on the ground and new deformation on buildings and vegetation. (2) Obvious scratch and displacement on the main scarp. (3)Cracks on the crown of landslide. | (1) Local deformation on the ground. (2) No obvious deformation on the main scarp. (3) No obvious expansion of the cracks on the buildings. (4) Small cracks on the crown of landslide. | (1) No sustained deformation on the ground. (2) No crack expansion on the landslide. And no new deformation on buildings and vegetation on the landslide. (3) No scratch and obvious displacement on the main scarp. |

Note: $F_s \neq 0$.

Fig. 2.

---

## Author Response (AR1)

**Response to the editor:**

Dear editor,

Thank you so much for working on our manuscript this hard pandemic time. Wish you healthy and everything smooth. Since we got the feedback, we worked very hard and revised the manuscript very carefully. The point-by-point response to the reviewers had been attached. Now we response the comments as following:

**Comments to the Author:**

Based upon the reviewers' comments, the manuscript needs substantial revisions. In detail, a revision of the English language is necessary, since in many parts the text is quite unclear and difficult to understand. Another crucial point is the use of such an approach in other landslide cases, as envisioned by the Authors: this should be better explained, in order to effectively evaluate the possibility in using such approach in other geological and morphological settings.

Authors are kindly invited to read carefully the reviewer's comments and prepare the revised manuscript accordingly.

Response: Thank you for your good comments. We checked the language very carefully. Dhruba Pikha Shrestha is one of the authors, who is the associate professor from ITC with good background of English writing. He has revised the relative sentences which are confusing or difficult to understand. Please see the sentences marked by blue color.

For another crucial point, we have added our explanation in the section of discussion in the manuscript. We specified the applicability of the approach and pointed out its limitation. We agree with you that the results from this study should be verified or tested in other landslide cases. Our research is based on detailed filed investigation, monitoring, and analysis in such specific landslides, we think the results should be applicable for the similar geological background areas prone to slow-moving landslides or similar landslide displacement process. We hope the results can be a good supplement for physical vulnerability of landslides. Currently, intensive researches on slow-moving landslides vulnerability in the Three-gorges Reservoir is strengthened, where we are applying our approach for more case studies. We are confident this approach can be verified and modified through our continuing studies. Essentially, the results of physical vulnerability of buildings on slow moving landslides are mostly related to the force of soils acting on building's foundation. Relatively, the quantitative relationship between the physical vulnerability of buildings and landslide displacement process is very weakly studied around the world. It needs more concentration of studies.

Sincerely yours,
Lixia Chen

**Point-by-point response to the reviews**

Dear Referee 1,

We would like to thank you for your professional and constructive comments concerning our manuscript entitled "Assessment of the physical vulnerability of buildings affected by slow-moving landslides". These comments are all valuable and helpful for revising and improving our manuscript. The main corrections in the manuscript and point-by -point to your comments are as following (the page number and line number in this refer to **the revised manuscript**).

**Specific comments:**

(1) In Section 2.2.1 the Authors recall the equivalent elastic beam – originally introduced by Burland and Wroth (1974) to define a damage ability criterion – in order to compute the maximum deflection exhibited by the same beam under a uniform load whose modulus equals q. In Figure 2 this uniform load acts horizontally, in correspondence of the lateral surface of the building's foundation affected by the landslide; whereas in Figure 3 the uniform load is applied vertically to the elastic beam. Accordingly, it is not clear in which direction the maximum deflection develops. Furthermore, symbols adopted in Figures 2 and 3 to denote the geometrical characteristics of the building's foundation are not internally consistent. Could the Authors better explain?

Response: thank you very much for the comment. In Figure 3, we actually try to express the uniform load applied horizontally, so sorry for the confusing. We plan to revise the figure 3 as follows.

[Figure]

**Fig.3. The simple beam with its foundation affected by landslide thrust.**

[Figure]

**Fig. 2. Schematic diagram of landslide thrust action on a building.**

We revised the symbols adopted in Figures 2 and 3 to ensure them consistent. Where $q$ denotes the distribution force on the foundation (kN/m), $F$ denotes the horizontal component of landslide residual thrust ($P_i$) in Eq. (3), and $h$ denotes the vertical distance from sliding surface to the ground surface. (Fig.2). $L$, $W$, and $d$ denote the length, width and depth of the building foundation (Fig.3).

(2) In the same Section 2.2.1 the concept of "inclination" of a building is introduced. Does this inclination corresponds to the "rotation" or "slope" (i.e. the change in gradient of a line joining two reference points of the foundation base) or to the "tilt" (describing the rigid body rotation of the whole superstructure or a well-defined part of it) defined by Burland and Wroth (1974)? Or does it refer to another well-defined parameter? Please clarify. I also suggest to associate the Eq. (9) – used to express mathematically the concept of inclination – with a Figure helpful to better understand the meaning of symbols adopted in Eq. (9), including the angle alpha.

Response: thank you very much for the suggestion. The incline angle of the building is the angle between the vertical plane of inclined building and the vertical plane of the original design building or the angle between the bottom plane of displacement foundation and the horizontal plane of foundation bottom of the original design. The inclination of the building is the tangent value of the incline angle. We add a figure to better understand the meaning the meaning of symbols adopted in Eq. (9) as follow. More explanation has also been put in the context. Please see line 150 to line 166.

[Figure]

**Fig. 4. The inclination of the building**

(3) In Table 3 the shear strength parameters of soils involved in the shear zone of the Manjiapo landslide are summarized. Are they residual shear strength values? And, more in general, what type of laboratory tests was carried out? Please explain.

Response: thank you very much for the comment. The shear strength parameters in Table 3 are residual values. According to the report provided by the China Geological Survey (Hunan Institute of Xiangxi Geological Engineering Survey) in 2017, six groups of undisturbed soil samples were collected from the shear zone of the Manjiapo Landslide. Obtained by residual shear tests in the laboratory, the shear strength parameters of slip soils in Table 3 are the average values of these six groups of soil samples. So we added more details about shear strength parameters in line 238 to line 240.

(4) In Table 4 it is not clear if the Young's modulus refers to the masonry constituting the building superstructure or to the material constituting the building foundation. Please clarify.

Responses: Thank you very much for your comments. The Young's modulus in Table 4 refers to the material constituting the building foundation. We will revise Table 4 as follows.

**Table 4. Parameters of the case building and its foundation on the Manjiapo landslide**

| For building | | | | For foundation | | | Soil depth where the building located ($m$) |
|---|---|---|---|---|---|---|---|
| Length $L$ ($m$) | Width $W$ ($m$) | Height $H$ ($m$) | Depth $d$ ($m$) | Young's modulus $E$ ($MPa$) | Shear modulus $G$ ($MPa$) | $E/G$ | |
| 25 | 9 | 2.8 | 1 | 2250 | 865 | 2.6 | 5 |

(5) In Section 4.1 the rainfall scenarios considered for transient seepage analyses are introduced. However, relevant information is provided neither on the fixed boundary conditions nor on the adopted hydraulic conductivities. Please improve this Section.

Response: thank you very much for the suggestion. We use the SEEP/W code (GEOSTUDIO) to analyze the groundwater seepage of Manjiapo landslide. We obtain the amount of 3-day cumulative precipitation corresponding to each return period by using PT (Pearson type) Ⅲ distribution model and the rainfall data (Fig. 12 in the revised manuscript). The average amount of 3-day cumulative precipitation is input to the software in turn, and then the groundwater under the rainfall scenarios is simulated.

The saturated volumetric water content is 0.4 by cutting ring method. The saturated permeability coefficient is obtained by back analysis. We choose the saturated volumetric water content and the permeability coefficient by the variable-controlling approach. Three groups of input values are: 0.4, 0.1; 0.4, 0.2; 0.4, 0.3. Then, the groundwater is simulated and then validated for the rainfall event in March 2018. The root mean square error (RMSE) is utilized to check the accuracy of calculation. Lower RMSE means smaller error and better prediction effect. The results of RMSE are shown in the following table. We find the saturated volumetric water content is 0.4 and the most suitable value of permeability coefficient is 0.3 m/d. We added the detailed analysis in the Section 4.1. Please see line 309 to line 320.

**Table 5. Permeability coefficient back analysis of the rainfall event in March 2018, by comparing the root mean square errors (RMSE) of for three hydrological gauges (installed by the authors in December 2017, see Fig.5) on Manjiapo landslide**

| The permeability coefficient (m/d) | 0.1 | 0.2 | 0.3 |
|---|---|---|---|
| RMSE (STK-1) | 2.280 | 2.222 | 2.154 |
| RMSE (STK-2) | 0.860 | 0.677 | 0.615 |
| RMSE (STK-3) | 2.540 | 2.491 | 2.405 |

Note: the saturated volumetric water content by Lab test is 0.4.

[Figure]

**Fig. 6. Geological profile of I-I' of the Manjiapo landslide (1:1 000).**

(6) In Table 5 the results obtained for the four considered rainfall scenarios are summarized. In all the cases, the factor of safety ($F_s$) is lower than 1. This would imply that the landslide is always moving, in disagreement with the information gathered by the Authors about the cracks on the ground surface of the Manjiapo landslide. In particular, the Authors observe that "Meanwhile, since the extreme rainfall events were recently rare, the deformation of the landslide did not obviously change, which was similar to the deformation situation in June 2016. For example, the cracks on the landslide did not expand, and the number of new cracks was very few" (page 10 – lines from 249 to 252). Probably, the shear strength parameters used for the limit equilibrium analyses are too low (see Table 3) and should be compared with those deriving from the back-analysis of the event occurred on June 2016.

Response: thank you very much for the suggestion. The factor of safety ($F_s$) in the original Table 5 is for the area where the building is located, but not for the whole landslide. This is pointed out by the sentence from line 195 to 197 on page 9. To avoid confusing, we revised this variable to be $F_{sb}$. In addition, we add the results of safety factors for the whole landslide under four scenarios in table 6 as follows. Now in this table, $F_s$ refers to the factor safety of the whole landslide. From table 6, we find that the landslide is stable with factor of safety 1.457.

**Table 6. Landslide residual thrust, pushing force on the building's foundation, and vulnerability of the building based on four scenarios: (a) dry condition without earthquake, (b) rainfall with a return period of 5 years (3-day precipitation 235 mm/d from Fig. 11), (c) rainfall with a return period of 10 years (3-day precipitation 279 mm/d from Fig. 11) without earthquake, and (d) rainfall with a return period of 50 years (3-day precipitation 352 mm/d from Fig. 11) without earthquakes.**

| Scenarios | $F_{sb}$ | $F_s$ | F(kN/m) | q(kN/m) | i(%) | V |
|---|---|---|---|---|---|---|
| a | 0.853 | 1.457 | 142 | 28 | 0.053 | 0.053 |
| b | 0.529 | 0.819 | 1756 | 351 | 0.656 | 0.656 |
| c | 0.481 | 0.778 | 2040 | 408 | 0.762 | 0.762 |
| d | 0.428 | 0.632 | 2638 | 528 | 0.985 | 0.985 |

Note: $F_{sb}$ is the safety factor of the area where the building is located.

(7) In Section 4.2 the results of the vulnerability analysis concerning a selected building within the affected area of the Manjiapo landslide are presented. Focusing on the obtained vulnerability curve (Fig. 12) the Authors observe that "the physical vulnerability is very low when the landslide is stable with a safety factor greater than 1.0" (lines from 326 to 327). How is this observation justifiable? Indeed, it is expected that the building vulnerability equals 0 (no damage) if the landslide does not move. In this regard, are the Authors sure that the chosen Weibull (1951) function is the best one to mathematically express the vulnerability curve when $1/F_S$ is adopted as landslide intensity parameter?

Response: thank you very much for the comment. For slow-moving landslides, they can have a $F_s$ greater than 1.0 but with cracks within the landslide area. Based on the Chinese standard of Specification of *Risk Assessment for Geological Hazard* (DZ/T 0286-2015), there are three stability states of landslide according to the range the safety factor ($F_s$) of landslide as the following table.

**Table 7. The range of safety factor (Fs) of landslide and its state (referred to Ministry of Land and Resources of the PRC, 2015)**

| The safety factor $F_s$ | $F_s \leqslant 1.00$ | $1.00 < F_s \leqslant F_{st}$ | $F_s > F_{st}$ |
|---|---|---|---|
| Stability state of landslide | unstable | Less stable | stable |
| Description | (1) Many newly expanded cracks on the ground and new deformation on buildings and vegetation. (2) Obvious scratch and displacement on the main scarp. (3) Cracks on the crown of landslide. | (1) Local deformation on the ground. (2) No obvious deformation on the main scarp. (3) No obvious expansion of the cracks on the buildings. (4) Small cracks on the crown of landslide. | (1) No sustained deformation on the ground. (2) No crack expansion on the landslide. And no new deformation on buildings and vegetation on the landslide. (3) No scratch and obvious displacement on the main scarp. |

Note: $F_s \neq 0$. And $F_{st}$ denotes the design safety factor which is defined according to the slope safety level and slope type.

When the whole landslide has $F_s$ value from 1.0 to $F_{st}$ (the design safety factor of the Manjiapo landslide is 1.30), the landslide will have small scale deformation or cracks. While the buildings located across the cracks can have damages with a certain degree.

In the Three Gorges Reservoir area, China (Chen et, 2016 ) and other areas, such as Moio della Civitella (Salerno province, Italy) (Infante et al., 2016), the buildings on the huge, slow-moving landslides will appear this state. So, to solve the problem on building's vulnerability, we need to focus on the local stability of this kind of landslide like Manjiapo landslide, but not the whole body. In Figure 13, the $F_s$ value is for the local stability of the soil where the case building located. Following the above question, we need to modify $F_s$ in Figure 13 to be $F_{sb}$. We can find from Figure 13 and Figure 16 that, when $1.0 < F_{sb} < 1.30$, the building vulnerability is from 0 to 0.1. This means the building is damaged very slightly, which is consistent with the real state of buildings on slow-moving landslides.

In this regard, we are sure that Weibull function is suitable to express the vulnerability curve. In fact, Weibull function is used to express the vulnerability curve in many present literatures and has high acceptance, such as Dario Peduto et (2017), Kang et (2016), Papathoma-Köhle (2016), Negulescu et (2010). We add the literatures in line 201 to 202.

Response: thank you very much for the comment. Table 2 expresses threshold values of the building inclination for three types of buildings with different height. In this manuscript, we focus on the first class of building with height lower than 24 m, which are common residential buildings in rural areas of China. As to this kind of building, the threshold is a fixed value 1%. Meanwhile, we use the vertical height of tilted building calculated from the outdoor ground to calculate the inclination of the case study building. Plsease see line 155 to 163.To avoid confusion, we will remove the parameter of the building height (from the outdoor ground) and don't make sensitivity analysis of the building height. We reselected some parameters to do sensitivity analysis. Please see line 376 to 378 and Figrue 14 and 15.

**Table 2. The threshold value of building inclination (Ministry of Housing and Urban–Rural Development of PRC, 2016).**

| Height $H_g$(m) | $H_g \leq 24$ | $24 < H_g \leq 60$ | $60 < H_g \leq 100$ |
|---|---|---|---|
| Threshold value $i_m$ | 1% | 0.7% | 0.5% |

Here, $H_g$ denotes the building height which is calculated from the outdoor ground.

(9) In my opinion, the vulnerability curves shown in Figure 15 have to be further validated before applying them in analyses at regional scale (lines from 403 to 408).

Response: thank you very much for the good comment. We are currently doing the researches on regional scale slow-moving landslide risk assessment in the Three Gorges reservoir area, China, which involves regional scale vulnerability assessment for buildings. We totally agree with you that before applying the results from this manuscript, we will do further validation. We rewritten the content in line 435 to 451.

**Technical corrections:**

(1) The symbol adopted throughout the manuscript to indicate the unit of measurement of force should be "kN" (lowercase k) instead of "KN".

Response: thank you very much for the good suggestion. We have modified "KN" to be "kN" in the revised manuscript.

(2)The symbol adopted in Table 3 to indicate the unit of measurement of stress should be "kPa" instead of "kpa".

Response: thank you very much for the good suggestion. We have modified "kpa" to be "kPa" in the revised manuscript.

(3) In the Note of Table 3 it is not clear if the information provided about the "permeability coefficient" and the "volume of water content" (or "volumetric water content"?) refers to the considered soil in saturated conditions or not. Relevant units of measurement should be provided, if applicable.

Response: thank you very much for the good suggestion. In the Note of Table 3, the information provided about the permeability coefficient and the volumetric water content refers to the considered soil in saturated conditions. The saturated volumetric water content is 0.4 by cutting ring method. The saturated permeability coefficient is obtained by back analysis. We find that the most suitable value of permeability coefficient is 0.3 m/d. We will add the detailed analysis in the Section 4.1.

Please see line 309 to line 320.

We tried our best to improve the manuscript and made some changes in the manuscript. We feel great thanks for your professional review work on our article, and hope that the correction and response will meet with approval.
Sincerely,
Lixia Chen

Dear Referee 2,

Thank you very much for your professional comments on our manuscript. These comments are all valuable and helpful for revising and improving our manuscript. The main corrections in the manuscript and the point-by-point responses to your comments are as following (the page number and line number in this letter refer to **the revised manuscript**):

**Specific Comments:**

(1) the Abstract should be rearranged and totally rewritten. It cannot be a list of steps followed during the analysis. Authors should mention the problem and the approach followed to get the results.

Response: Thank you for your good comments. We have rearranged and rewritten the abstract, which mentioned the problem and the approach followed to get the results. Please see line 11 to 26.

(2) Fig.3, pag.6. Authors should better clarify, for instance with an additional Figure, how the lateral forces impacting the foundation can be associated with $y_m$ (that is the inflection under vertical loads). At the moment the concept of $i_m$ is not clear.

Response: Thank you for your good comments. Figure 3 on page 6 did not clearly express the direction of lateral forces impacting the foundation. We try to express the uniform load applied horizontally, so sorry for the confusing. We will modify this figure as follows.

[Figure]

**Fig.3. The simple beam with its foundation affected by landslide thrust.**

The concept of $i_m$ is the threshold value of inclination of buildings. Buildings with inclination exceeding $i_m$ are dangerous and uninhabitable. In Table 2, we listed out the standard of threshold values for three kinds of buildings with different height. Plsease see line 183 to 184.

(3)The sentence (pag.7) referring to Finno et al. (2005) should be better clarified.

Response: Thank you for your good comments. The original sentence is: Since cracks on walls are not visible, especially when the building with high stiffness is exceedingly inclined because of the ground deformation, they usually serve as the indicators of damage degree evaluation if the building stiffness is small (Finno et al., 2005)

Sorry for the confused expression in the above sentence. We want to clarify that cracks on building walls are not the only indicator to assess vulnerability, especially when the building has a very good stiffness. So, we revised this sentence as follows.

Finno et al. (2005) reported that when highly stiff buildings are very inclined due to ground deformation, the wall cracking phenomenon is not obvious. On the contrary, if the stiffness of the building is lower, the cracking on the wall becomes serious. This research shows that using only cracks as an indicator is not suitable for vulnerability assessment. Other indicators, such as inclination, should also be taken into consideration. We revised the English sentence in line 175 to 180.

(4) The Authors identify the damage classification with vulnerability. This aspect deserves further clarifications based on widely shared literature.

Response: Thank you for your good comments. In order to simplify the research work, many researchers directly use damage degree as vulnerability. Tarbotton et al. (2015) defined empirical vulnerability functions as "a continuous curve associating the intensity of the hazard (X-axis) to the damage response of a building (Y-axis)". Kang et al. (2016) think that the range of damage to the buildings makes it possible to assess the vulnerability using a vulnerability curve that relates the intensity of debris flow with the degree of damage. They use the degree of damage to the buildings to estimate vulnerability. We added the literature in line 172 to 173.

**Table 7. The range of safety factor (*Fs*) of landslide and its state (referred to Ministry of Land and Resources of the PRC, 2015)**

| The safety factor $F_s$ | $F_s \leqslant 1.00$ | $1.00 < F_s \leqslant F_{st}$ | $F_s > F_{st}$ |
|---|---|---|---|
| Stability state of landslide | unstable | Less stable | stable |
| Description | (1) Many newly expanded | (1) Local deformation on the | (1) No sustained deformation on |

| | | |
|---|---|---|
| cracks on the ground and new deformation on buildings and vegetation. (2) Obvious scratch and displacement on the main scarp. (3) Cracks on the crown of landslide. | ground. (2) No obvious deformation on the main scarp. (3) No obvious expansion of the cracks on the buildings. (4) Small cracks on the crown of landslide. | the ground. (2) No crack expansion on the landslide. And no new deformation on buildings and vegetation on the landslide. (3) No scratch and obvious displacement on the main scarp. |

Note: $F_s \neq 0$. $F_{st}$ denotes the design safety factor.

The value of $F_{st}$ is defined according to the slope safety level and slope type (Table 8). Meanwhile the slope safety level is defined based on the potential economic loss and element at risk. According to the field investigation, there are 116 residents in the affected area of the Manjiapo landslide, and the road passes through the middle part of the landslide. In case of geologic hazard, it will threaten the lives and property of 116 residents and damage more than 67,000 square meters of the land. At the same time, the road will be damaged, threatening the safety of the pedestrians and passing vehicles. The potential economic loss will be more than CNY 5 million. the safety level of the Manjiapo landslide is judged to be second level based on below table.

**Table 8. The value of the design safety factor (referred to Ministry of Housing and Urban–Rural Development of PRC, 2013)**

| the slope safety level | | First level | Second level | Third Level |
|---|---|---|---|---|
| Permanent slope | General condition | 1.35 | 1.30 | 1.25 |
| | Earthquake condition | 1.15 | 1.10 | 1.05 |
| Temporary slope | | 1.25 | 1.20 | 1.15 |

**Table 9. The slope safety level (referred to General Administration of Quality Supervision, Inspection and Quarantine of the PRC,2016)**

| the slope safety level | | First level | Second level | Third Level |
|---|---|---|---|---|
| potential economic loss (CNY) | | $\geq 50$ million | 5 million to 50 million | < 5million |
| Element at risk | population | $\geq 500$ | 100 to 500 | < 100 |
| | Infrastructure | Very important | Important | less important |

Note: If one of the conditions is met, it will be judged to be the corresponding slope safety level.

So, when the safety factory of the Manjiapo Landslide is greater than 1.30, the landslide is stable and the landslide intensity is very low. In addition, the resistance ability of the building can prevent the building from being destroyed by the low intensity of the landslide (Du, 2013). In summary, the physical vulnerability of the building on Manjiapo landslide is very low when the safety factory is greater than 1.30. It provides that the physical vulnerability of the building on Manjiapo landslide is 0 when the reciprocal value of the safety factor is 0.5. The physical vulnerability of the case study building on Manjiapo landslide is demonstrated in Fig. 13. We add more details to explain the range of variation of $1/F_s$ in line 344 to 365.

[Figure]

**Fig. 13. The physical vulnerability curve for masonry buildings impacted by the slow-moving landslides.**

(7) The comparison shown in Figure 13 is unclear. What is there on x-axis?

Response: thank you very much for the comment. By putting these parameters together on a single diagram, we can clearly compare and find out which parameter is more sensitive to vulnerability. We modified the figure. In the new figure, the X-axis stands for the variability rate of the parameter.

[Figure]

**Fig .15. The sensitivity analysis of building parameters for physical vulnerability.**

We can find that the length of the building has the most significant influence on the physical vulnerability of building and the width of the building is the second major factor. So, we modify the figure 16 to show the physical vulnerability curves of building with different length and width. We modified the content in line 375 to 378 and line 389 to 390 and line 403 to 408 and Table 10.

[Figure]

**Fig. 16. Physical vulnerability curves of buildings with different parameters: (a) length and (b) width.**

(8) Conclusions: please clarify better or add references concerning the calculation of $F_S$ just in correspondence of buildings and over large areas. - Exportability should be better supported with clarifications.

Response: thank you very much for the suggestion. We are currently doing the researches on regional scale slow-moving landslide risk assessment in the Three Gorges reservoir area, China, which involves regional scale vulnerability assessment for buildings. The topic in this manuscript is partially new. There are rare references presently concerning $Fs$ of slow-moving landslides and vulnerability of buildings. But the researches about calculation of $Fs$ over large areas can be found from some researches, such as Muntohar AS, Liao HJ (2009), Apip, Takara K, Yamashiki Y, et al (2010), Salciarini (2006) and Sorbino (2010). We are eager to link the intensity of slow-moving landslides with vulnerability of buildings over large areas. Before applying the results from this manuscript, we will do further validation. We revised some content in discussion and conclusions. Please see line 435 to 453 and line 461 to 467.

We tried our best to improve the manuscript and made changes in the manuscript. We feel great thanks for your professional review work on our article, and hope that the responses will meet with approval.

Sincerely,
Lixia Chen

[revised manuscript text omitted]

---

## Author Response (AR2)

**Response to the editor:**

Dear editor,

Thank you so much for working on our manuscript this hard pandemic time. Wish you healthy and everything smooth. Since we got the feedback, we revised the manuscript very carefully following the indications by the reviewers. The point-by-point response to the reviewers had been attached.

Sincerely yours,

Lixia Chen

**Point-by-point response to the reviews**

Dear professor,

We would like to thank you for your professional and constructive comments concerning our manuscript entitled "Assessment of the physical vulnerability of buildings affected by slow-moving landslides". These comments are all valuable and helpful for revising and improving our manuscript. The main corrections in the manuscript and point-by -point to your comments are as following (the page number and line number in this refer to **the revised manuscript**).

**Specific comments:**

(1) Page 8 – Equation (8). I would observe that the maximum deflection of a simple rectangular beam as expressed with the Equation (8) corresponds to the maximum horizontal displacement of the foundation (see Page 8 - Lines from 165 to 166) only if the edges of the beam (or the foundation) are fixed. This is a work hypothesis that, in my opinion, has to be highlighted.

Response: thank you very much for your good suggestion. We added the work hypothesis. Please see line 162 to 163.

(2) Page 8 – line 164. The authors claim that "It is worth pointing out that the building in our study case is regarded as a rigid building". This is not clear for me. Reading the paper I understood that superstructure deforms as the foundation deforms due to the action exerted by the landslide thrust. Please clarify.

Response: thank you very much for the comment. When the building is regarded as a rigid body, it means that the displacement of foundation is the same as the displacement of the superstructure. So, we can use the ratio of the horizontal displacement of the foundation ($y_m$) to the vertical height of tilted building ($H$) as the inclination ratio of the building.

(3) Page 9 – Lines 196 to 197. The authors observe that "It is important to note that FS is calculated for only the area where the building under study is located, but not for the whole landslide area". This is not clear for me. Did the authors consider different slip surfaces in applying the Limit Equilibrium Method explained in Section 2.1? In the case of a positive answer, are the soil shear strength parameters the same (i.e. the ones summarized in Table 3) for all the considered slip surfaces?

Response: thank you very much for the comment, and we are sorry for the confusing. There is only one slip surface for this landslide. We originally want to clarify that we specially calculated the local $F_s$ for the area where buildings located (it is lately expressed as $F_{sb}$), besides the $F_s$ calculation for the whole landslide. The reason why we focus on the local stability of the soil is that for slow-moving landslides, they can have a $F_s$ greater than 1.0 but with cracks within the landslide area, which can cause damages on buildings located across the cracks. This kind of cases can also be found in the Three Gorges Reservoir area, China (Chen et, 2016 ) and other areas, such as Moio della Civitella (Salerno province, Italy) (Infante et al., 2016). So, to solve the problem on building's vulnerability, we need to focus on the local stability of this kind of landslide like Manjiapo landslide, not only the whole body. We have revised the sentences and added more to clarify the meaning, please see line 194 to line 199 and and line 370 to line 371.

Response: thank you very much for the good comment. We obtained the different vulnerability curves by controlling the variables. In Fig.16a, the building width is fixed as W=9m when testing four building lengths: 15 m, 20 m, 25 m, and 30 m. Then we can observe four types of building with the ratio of 15/9, 20/9, 25/9, and 30/9. When the ratio is closed to 1, the vulnerability of the building is lower than the other cases under the same value of $F_{sb}$. To better explain the sentence, we revised the figure 16 (a and b) by adding the appointed width or length. Please see the fig.16 and line 405 to 408.

[Figure]

**Fig. 16. Physical vulnerability curves of buildings with different parameters: (a) length and (b) width.**

Technical corrections

(1) Page 7 – line 145. In the text, one can read: "When = L/2". It should be corrected as: "When x = L/2".

Response: thank you very much for your suggestion. We have revised it. Please see line 145.

(2) Page 7 – line 151. Since using the terms "vertical plane" twice in the same sentence may be misleading, I would suggest rephrasing the sentence.

Response: thank you very much for your suggestion. We have revised the sentence. Please see line 150 to line 151.

We tried our best to improve the manuscript and made some changes in the manuscript. We feel great thanks for your professional review work on our article, and hope that the correction and response will meet with approval.

Sincerely,
Lixia Chen

[revised manuscript text omitted]